# TriQDef: Disrupting Semantic and Gradient Alignment to Prevent Adversarial Patch Transferability in Quantized Neural Networks

**Amira Guesmi**
eBrain Lab
Engineering Department
New York University Abu Dhabi
Abu Dhabi, UAE

**Bassem Ouni**
Khalifa University
Abu Dhabi, UAE

**Muhammad Shafique**
eBrain Lab
Engineering Department
New York University Abu Dhabi
Abu Dhabi, UAE

## Abstract

Quantized Neural Networks (QNNs) are widely deployed in edge and resource-constrained environments for their efficiency in computation and memory. While quantization distorts gradient landscapes and weakens pixel-level attacks, it offers limited robustness against patch-based adversarial attacks—localized, high-saliency perturbations that remain highly transferable across bit-widths. Existing defenses either overfit to specific quantization settings or fail to address this cross-bit vulnerability. We propose **TriQDef**, a tri-level quantization-aware defense framework that disrupts the transferability of patch-based attacks across QNNs. TriQDef integrates: (1) a *Feature Disalignment Penalty (FDP)* that enforces semantic inconsistency by penalizing perceptual similarity in intermediate features; (2) a *Gradient Perceptual Dissonance Penalty (GPDP)* that misaligns input gradients across quantization levels using structural metrics such as Edge IoU and HOG Cosine; and (3) a *Joint Quantization-Aware Training Protocol* that applies these penalties within a *shared backbone* jointly optimized across multiple quantizers. Extensive experiments on CIFAR-10 and ImageNet show that TriQDef lowers Attack Success Rates (ASR) by over 40% on unseen patch and quantization combinations while preserving high clean accuracy. These results highlight the importance of disrupting both semantic and perceptual gradient alignment to mitigate patch transferability in QNNs.

## 1 Introduction & Related Work

Quantized Neural Networks (QNNs) offer a compelling trade-off by significantly reducing memory and compute requirements while maintaining competitive accuracy (Liu et al., 2021a; Katare et al., 2023; Zhang & Chung, 2021; Tonellotto et al., 2021; Hernández et al., 2024). Prior studies have shown that quantization can distort gradient landscapes, thereby weakening the effectiveness of traditional pixel-level adversarial attacks (Li et al., 2024; Yang et al., 2024). However, such gradient masking effects offer little protection against more structured threats.

*Adversarial patch attacks* (Karmon et al., 2018; Brown et al., 2017; Chen et al., 2022; Guesmi et al., 2024) pose a unique challenge by inserting localized, high-saliency patterns that hijack model predictions. Unlike subtle pixel perturbations, these patches are robust to input variation and generalize across architectures and quantization levels Guesmi et al. (2023). Crucially, our analysis reveals that even under aggressive quantization (e.g., 2-bit), adversarial patches crafted on full-precision models maintain high success rates, highlighting a critical blind spot in current quantization-aware defenses.

Existing defenses such as Projected Gradient Descent (PGD)-based adversarial training (Madry et al., 2018) offer limited effectiveness against patch attacks, which exploit model attention rather than gradient sensitivity. Classical input-transformation defenses including MagNet (Meng & Chen, 2017), Feature Squeezing (Xu et al., 2017), and randomized input transformations as explored by

Tramèr et al. (Tramèr et al., 2017a) provide robustness against pixel-level perturbations but are substantially less effective against large, structured adversarial patches that bypass small-noise assumptions. Patch-Based Adversarial Training (PBAT) (Rao et al., 2020) incorporates patch patterns during training and improves robustness on seen configurations, but often fails to generalize across novel patch types or bit-widths. Other approaches, such as Double-Win Quantization (DWQ) (Fu et al., 2021), stochastic precision inference (Sen et al., 2020), and feature-space smoothing (Song et al., 2020), primarily target pixel-level noise and do not directly tackle the structured, cross-bit nature of adversarial patches (Xiao et al., 2023). Input preprocessing-based defenses such as (Tarchoun et al., 2023; Nie et al., 2022) impose significant computational overhead, undermining the efficiency gains that quantization aims to provide.

In this work, we introduce **TriQDef**, a unified defense framework designed to explicitly disrupt the core enablers of patch-based adversarial transferability in QNNs. Our analysis reveals that quantized models (even under extreme bit-width reduction) exhibit surprisingly high vulnerability to transferable adversarial patches. This phenomenon arises from persistent alignment in both internal features and input gradient signals across bit-widths. TriQDef addresses this vulnerability through three synergistic components that target semantic and optimization-level consistency:

**Feature Disalignment Penalty (FDP)** enforces semantic divergence by penalizing perceptual similarity in feature maps across quantized variants. Using differentiable variants of Edge IoU and HOG Cosine similarity, FDP encourages each bit-width to develop unique feature representations, thus weakening patch generalization. **Gradient Perceptual Dissonance Penalty (GPDP)** misaligns the saliency landscape by penalizing structural and perceptual gradient similarity across bit-widths. It directly targets gradient-level alignment that facilitates adversarial transfer, extending beyond cosine similarity to include perceptual alignment in edge structures and gradient orientations. **Bit-Width-Aware Curriculum Training (BACT)** stabilizes optimization by staging the activation of quantizers on the same backbone $\theta$: training begins at higher precision and progressively enables lower-bit $Q_b$; losses are computed over the active set $\mathcal{B}_t$ so that $L_{\text{FDP}}$ and $L_{\text{GPDP}}$ act across bit-widths.

Our main contributions are summarized as follows:

- We conduct, to our knowledge, the first systematic study of patch-based transferability in QNNs, demonstrating that adversarial patches remain highly effective across quantization levels, including 2-bit regimes.

- We propose **TriQDef**, a tri-component defense targeting both feature and gradient alignment, explicitly designed to prevent cross-bit patch generalization.

- We introduce perceptual alignment metrics (Edge IoU and HOG Cosine Similarity) as theoretically justified tools to quantify and disrupt semantic and gradient-level alignment across bit-widths. These metrics provide a principled alternative to cosine similarity by capturing structural and textural alignment that underlies patch transferability in QNNs.

- Our approach reduces attack success rate (ASR) by over 40% on unseen patch and quantization configurations across CIFAR-10 and ImageNet, outperforming PBAT and DWQ with around 2% drop in clean accuracy.

- Ablation studies validate the complementary role of each module and reveal that quantization alone does not sufficiently alter the shared attack surface, highlighting the need for targeted perceptual and structural misalignment.

**TriQDef** challenges the assumption that quantization inherently enhances adversarial robustness. By explicitly dismantling shared vulnerabilities at both the representational and gradient levels, TriQDef provides a principled and extensible framework for securing QNNs against patch-based threats.

## 2 MOTIVATION

Despite recent progress in adversarial training and quantization-aware techniques, we show that QNNs remain highly vulnerable to structured, localized adversarial attacks, particularly adversarial patches. This vulnerability stems from a critical oversight: existing defenses do not generalize across quantization levels, and thus fail to prevent cross-bit transferability of patch-based attacks.

Our investigation reveals two key limitations that motivate the need for a principled, quantization-aware patch defense framework.

**Adversarial Patches Transfer Effectively Across Bit-Widths.** We begin by evaluating the transferability of adversarial patches crafted on full-precision (32-bit) models to quantized models trained using Quantization-Aware Training (QAT). Table 1 reports the Attack Success Rates (ASR) of two state-of-the-art patch attacks (LAVAN (Karmon et al., 2018) and GAP (Brown et al., 2017)) on various QNN architectures. Notably, the adversarial patches retain high effectiveness even under extreme quantization (e.g., over 73% ASR on 2-bit ResNet-56), confirming that quantization alone offers limited resilience against structured perturbations. This cross-bit vulnerability persists despite the reduced numerical precision and quantization noise introduced by QAT.

**Standard and Patch-Based Adversarial Training Fail to Generalize Across Bit-Widths.** We further analyze whether existing adversarial defense methods can mitigate this vulnerability. In Table 2, we compare standard adversarial training (AT) and Patch-Based Adversarial Training (PBAT) under different quantization paradigms: full-precision (FP), 8-bit Quantization-Aware Training (QAT), and 8-bit Post-Training Quantization (PTQ). While

Table 1: ASR (%) of LAVAN and GAP (6x6 patches) transferred from full-precision models to QAT-trained QNNs on CIFAR-10.

| Attack | LAVAN | | | | GAP | | | |
|--------|-------|------|------|------|-------|------|------|------|
| Model | 32bit | 8bit | 4bit | 2bit | 32bit | 8bit | 4bit | 2bit |
| Res-56 | 86.43 | 83.24 | 76.22 | 73.08 | 84.40 | 56.69 | 54.22 | 47.91 |
| Res-20 | 87.22 | 83.73 | 77.30 | 74.18 | 84.71 | 59.61 | 58.45 | 50.31 |
| VGG-19 | 88.95 | 85.56 | 79.81 | 77.19 | 95.79 | 59.65 | 48.70 | 40.69 |
| VGG-16 | 87.17 | 84.73 | 78.29 | 76.67 | 95.71 | 64.24 | 52.04 | 48.90 |

PBAT significantly reduces ASR for patch types seen during training, its robustness deteriorates sharply on unseen patch configurations—particularly when the patch was generated or tested under a different bit-width. For instance, ASR increases by more than 20% when evaluated on 2-bit patches not seen during training. These results reveal a failure to generalize across quantization shifts, underscoring the need for defenses that explicitly target bit-level adversarial generalization.

To further support these observations, we present in Appendix A: (1) cross-architecture transfer results including vision transformers; (2) analysis under dynamic and post-training quantization; (3) results using additional patch-based attacks (e.g., DPR, PatchAttack); and (4) comprehensive ablations and visualizations. These collectively highlight the limitations of existing defenses and motivate the design of **TriQDef**—a tri-level framework that breaks patch transferability via semantic, gradient, and curriculum-based alignment disruption.

Table 2: ASR (%) of LAVAN attack across training paradigms and quantization levels. PBAT-trained models fail to generalize to unseen patch bit-widths.

| Patch Type | Standard | | | PBAT | | |
|------------|----------|-----|-----|------|-----|-----|
| | FP | QAT | PTQ | FP | QAT | PTQ |
| 8×8 (Seen) | 88.17 | 81.56 | 85.24 | 40.39 | 40.56 | 45.44 |
| 10×10 (Seen) | 92.33 | 84.33 | 87.48 | 57.86 | 56.77 | 60.60 |
| 8×8 (Unseen, 4-bit) | 89.92 | 83.40 | 86.78 | 62.10 | 71.42 | 75.16 |
| 10×10 (Unseen, 2-bit) | 91.18 | 85.62 | 87.91 | 65.30 | 78.34 | 81.09 |

## 3 METHODOLOGY

### 3.1 OVERVIEW

We propose **TriQDef**, a framework that mitigates the transferability of patch-based adversarial attacks in QNNs. TriQDef integrates three complementary components into a cohesive training strategy. The first component, *Feature Disalignment Penalty (FDP)*, disrupts semantic consistency by encouraging divergence in internal feature representations across different quantization levels. The second component, *Gradient Perceptual Dissonance Penalty (GPDP)*, penalizes perceptual alignment in input gradients between bit-widths, targeting edge- and texture-level similarity that facilitates cross-bit transferability. The third component, *Bit-Width-Aware Curriculum Training (BACT)*, stabilizes training under extreme quantization by progressively *enabling* lower-bit quantizers on the same backbone, starting from higher precision and expanding the active bit-width set over time.

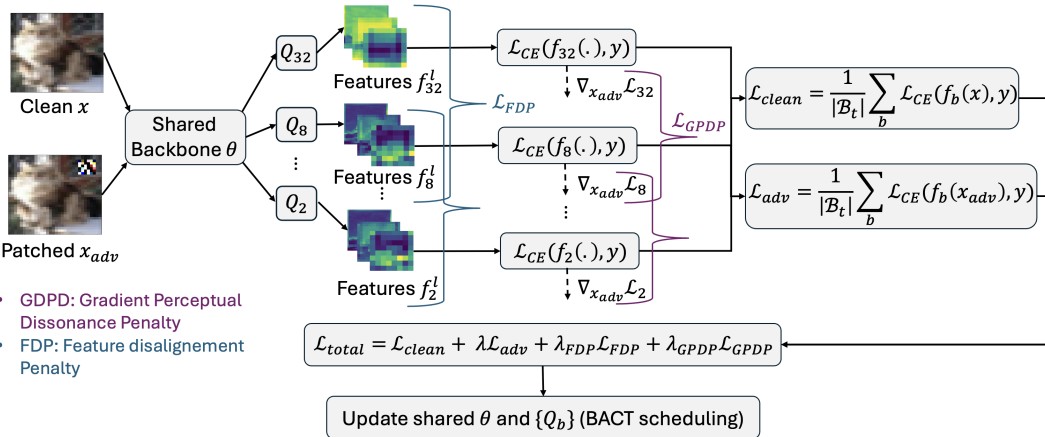

Figure 1: TriQDef overview. A single shared backbone $\theta$ is paired with multiple quantizers $\{Q_b\}$ (e.g., 32/8/2-bit). Clean and adversarial inputs produce bit-specific views whose intermediate features (for $L_{\text{FDP}}$) and input gradients (for $L_{\text{GPDP}}$) are contrasted across bit-widths. Losses are aggregated into $L_{\text{total}}$ and used to update $\theta$ and $\{Q_b\}$ under a BACT schedule. Inference uses a single forward pass with the deployed $Q_b$ (no runtime overhead).

## 3.2 FEATURE DISALIGNMENT PENALTY (FDP)

As we show in Section 2, adversarial patches remain highly transferable across quantized neural network (QNN) variants, despite the reduced numerical precision. We argue that this transferability is facilitated by a phenomenon we call *semantic alignment across bit-widths*—where internal representations across quantized models remain perceptually similar even under adversarial attack.

To assess this, we quantify perceptual similarity using two established descriptors: (i) **edge-based overlap**, computed using the Sobel operator and Intersection-over-Union (IoU) (Zhang et al., 2018), and (ii) **textural similarity**, captured using Histogram of Oriented Gradients (HOG) (Dalal & Triggs, 2005), a robust descriptor widely used in computer vision and feature analysis. Figure 2 illustrates this behavior. Using a ResNet model on ImageNet, we analyze feature maps from clean and patched inputs across different bit-widths (full precision (fp), 5bit, 4bit, 2bit) and multiple layers. We compute pairwise Edge IoU and HOG Cosine Similarity as perceptual proxies. Results averaged over 100 samples reveal consistently high similarity, especially between adjacent quantization levels (e.g., 5bit $\leftrightarrow$ 4bit), indicating strong structural alignment that supports cross-bit patch generalization. To mitigate this, we introduce the **Feature Disalignment Penalty (FDP)**, a regularizer inspired by perceptual similarity metrics and soft alignment losses (Zhang et al., 2018). Unlike defenses that learn patch-specific filters, FDP explicitly discourages feature alignment across quantized variants by penalizing structural and textural similarity at selected intermediate layers during training. Let $f_b^{(l)}(x_{\text{adv}})$ denote the activation at layer $l \in \mathcal{L}$ of model $f_b$ with bit-width $b$, given a patched input $x_{\text{adv}}$. FDP measures both structural and textural alignment between models $b_i \neq b_j$ using two

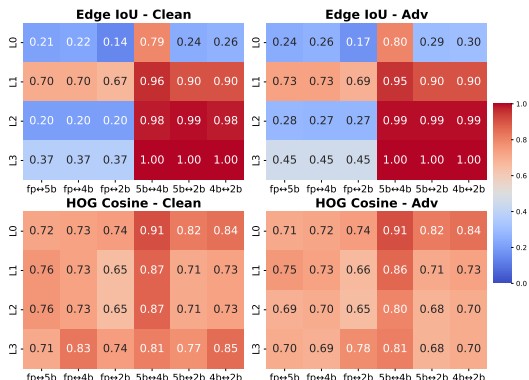

Figure 2: **Perceptual Alignment Across Bit-Widths.** Heatmaps show pairwise **Edge IoU** and **HOG Cosine Similarity** across feature maps extracted at four convolutional layers in (L0–L3) from ResNet variants (fp, 5bit, 4bit, 2bit). High similarity persists even under adversarial input, enabling patch transferability.

perceptual metrics: **Edge IoU**: Intersection-over-Union between binarized Sobel edge maps of $f_b^{(l)}$. **HOG Cosine Similarity:** Cosine similarity between Histogram of Oriented Gradient (HOG) descriptors.

The FDP loss is defined as:

$$\mathcal{L}_{\text{FDP}} = \sum_{l \in \mathcal{L}} \sum_{\substack{b_i, b_j \in \mathcal{B} \\ b_i \neq b_j}} \left[ \alpha \cdot \text{IoU}\big(\mathcal{E}(f_{b_i}^{(l)}(x_{\text{adv}})), \mathcal{E}(f_{b_j}^{(l)}(x_{\text{adv}}))\big) + \beta \cdot \cos\big(\phi(f_{b_i}^{(l)}(x_{\text{adv}})), \phi(f_{b_j}^{(l)}(x_{\text{adv}}))\big) \right] \quad (1)$$

Because traditional perceptual metrics (e.g., hard thresholded edges, non-differentiable HOG) are not suitable for gradient-based optimization, we adapt recent differentiable approximations such as SoftDice (Sudre et al., 2017) and smooth HOG descriptors (Kachouane et al., 2012) to create an end-to-end trainable loss.

$$\mathcal{L}_{\text{FDP}} = \sum_{l \in \mathcal{L}} \sum_{\substack{b_i, b_j \in \mathcal{B} \\ b_i \neq b_j}} \left[ \alpha \cdot \text{SoftDice}\left( S\left( E\left( f_{b_i}^{(l)}(x_{\text{adv}}) \right) \right), S\left( E\left( f_{b_j}^{(l)}(x_{\text{adv}}) \right) \right) \right) \right.$$
$$\left. + \beta \cdot \cos\left( H\left( f_{b_i}^{(l)}(x_{\text{adv}}) \right), H\left( f_{b_j}^{(l)}(x_{\text{adv}}) \right) \right) \right] \quad (2)$$

where $E(\cdot)$ computes Sobel edge magnitudes over spatial dimensions, $S(\cdot)$ applies soft binarization using a sigmoid with quantile-based threshold: $S(A; \tau, k) = \sigma\left( k \cdot (A - \tau) \right), \quad \tau = \text{quantile}(A, q)$, with sharpness $k = 100$, percentile $q = 85$. $\text{SoftDice}(A, B) = \frac{2 \cdot \sum A \cdot B}{\sum A + \sum B + \epsilon}$. $H(\cdot)$ computes a normalized HOG descriptor with $4 \times 4$ pixels per cell and $2 \times 2$ cells per block. $\alpha = 0.5$, $\beta = 1.0$ weighting hyperparameters.

The choice of hyperparameters is based on ablation studies on their sensitivity (see Appendix C).

**Why Use HOG Cosine Similarity?** HOG captures local edge orientation distributions and is robust to minor quantization noise and scale distortions. We observe high HOG similarity across bit-widths—indicating preserved structure—even when raw cosine similarity or Edge IoU degrade. This makes HOG a strong candidate for identifying transferable perceptual patterns.

**Why Use Edge-based SoftDice?** Edge IoU offers structural insight, but hard binarization is non-differentiable. SoftDice over softly binarized edges ensures smooth gradients while preserving interpretability. It captures shape-level alignment that HOG may overlook.

**Why Not LPIPS?** LPIPS, while widely used for perceptual similarity, is less suitable in our setting for several reasons. It is optimized for high-level semantic similarity based on human visual perception, making it less sensitive to the structural and directional patterns found in gradient maps or early-layer features that underlie patch transferability in quantized models. Moreover, LPIPS requires three-channel, large-resolution inputs and cannot be directly applied to single-channel or low-resolution feature maps or gradients. In contrast, Edge IoU and HOG Cosine Similarity offer lightweight, interpretable, and structurally grounded measures. They effectively capture spatial alignment (edges) and texture/orientation similarity (HOG) in both features and gradients, making them more appropriate for quantifying and disrupting perceptual alignment across bit-widths.

---

**Algorithm 1:** Compact FDP Training with Soft Disalignment

**Input** : Quantized models $\{f_b\}$, adversarial patch $P$, batch $(x, y)$, target layers $\mathcal{L}$, mask $M$

**Hyperparameters** : $\lambda_{\text{FDP}}, \alpha, \beta$

1 $x_{\text{adv}} \leftarrow x \odot (1 - M) + P \odot M$
2 **foreach** $b_i \in \mathcal{B}, l \in \mathcal{L}$ **do**
3 $\quad f_{b_i}^{(l)} \leftarrow f_{b_i}^{(l)}(x_{\text{adv}})$
4 $\mathcal{L}_{\text{FDP}} \leftarrow 0$
5 **foreach** $l \in \mathcal{L}, (b_i, b_j), i \neq j$ **do**
6 $\quad E_i \leftarrow \text{Sobel}(\text{mean}(f_{b_i}^{(l)}))$,
$\quad E_j \leftarrow \text{Sobel}(\text{mean}(f_{b_j}^{(l)}))$
7 $\quad H_i \leftarrow \text{SoftHOG}(f_{b_i}^{(l)}), \quad H_j \leftarrow \text{SoftHOG}(f_{b_j}^{(l)})$
8 $\quad \mathcal{L}_{\text{FDP}} += \alpha \cdot \text{SoftDice}(E_i, E_j) + \beta \cdot \cos(H_i, H_j)$

---

**Why FDP Works?** By introducing bit-level perceptual disalignment at key layers, FDP disrupts shared internal cues that adversarial patches exploit. This effectively breaks the cross-bit representational invariance that makes patches transferable. This perceptual disalignment strategy complements adversarial training by targeting a root enabler of transferability—shared structure across models—which remains underexplored in the literature on quantized model robustness (Li et al.,

2024). We provide a theoretical motivation for the FDP in Appendix B.1, grounding its design in principles of perceptual alignment, representation similarity, and adversarial vulnerability across quantized models.

**Training with FDP.** Algorithm 1 outlines our training procedure with the Feature Disalignment Penalty. For each batch, we apply an adversarial patch and extract intermediate features from multiple quantized models at selected layers. We then compute pairwise perceptual similarity using SoftDice over edge maps and cosine similarity over differentiable HOG descriptors. The total loss combines the standard cross-entropy with the FDP regularizer, guiding the models to develop divergent internal representations and weaken cross-bit adversarial transferability.

### 3.3 GRADIENT PERCEPTUAL DISSONANCE PENALTY (GPDP)

As shown in Table 3, although gradient *cosine similarity (CS)* is low across quantized models, suggesting directional disagreement, we observe persistently high perceptual similarity in the gradient maps. In particular, gradients exhibit strong *HOG Cosine Similarity* and moderate *Edge IoU*, revealing a hidden perceptual alignment that traditional CS fails to capture. We posit that this alignment facilitates the transferability of adversarial patches between bit-width variants by preserving texture and edge structure in gradient saliency.

To address this, we propose the **Gradient Perceptual Dissonance Penalty (GPDP)**, a perceptual regularizer designed to break gradient alignment across quantized models. GPDP penalizes both structural (edge-based) and textural (orientation-based) similarity in gradients, promoting gradient diversity that weakens the transferability of patch-based attacks. Let $\nabla_x^{b_i}$ be the input gradient from a model quantized to bit-width $b_i$. We define the GPDP loss as:

Table 3: Gradient similarity across bit-width models using different metrics. Despite low cosine similarity, perceptual metrics (HOG Cosine and Edge IoU) reveal strong structural alignment.

| Metric | fp↔5b | fp↔4b | fp↔2b | 5b↔4b | 5b↔2b | 4b↔2b |
|---|---|---|---|---|---|---|
| Cosine Sim. | 0.05 | 0.06 | 0.05 | 0.25 | 0.10 | 0.13 |
| Edge IoU | 0.14 | 0.15 | 0.14 | 0.20 | 0.15 | 0.15 |
| HOG CS | 0.81 | 0.81 | 0.80 | 0.82 | 0.81 | 0.81 |

$$\mathcal{L}_{\text{GPDP}} = \sum_{\substack{b_i, b_j \in \mathcal{B} \\ b_i \neq b_j}} \left[ \alpha \cdot \text{SoftDice}\big(\text{Sobel}(\nabla_x^{b_i}), \text{Sobel}(\nabla_x^{b_j})\big) + \beta \cdot \cos\big(\text{SoftHOG}(\nabla_x^{b_i}), \text{SoftHOG}(\nabla_x^{b_j})\big) \right] \quad (3)$$

Here, $\text{Sobel}(\cdot)$ computes edge maps from the gradient, and $\text{SoftHOG}(\cdot)$ is a differentiable version of the Histogram of Oriented Gradients (HOG) descriptor. SoftDice measures structural overlap, while the cosine of SoftHOG descriptors captures perceptual similarity. Coefficients $\alpha$ and $\beta$ balance these two components, with values set to $\alpha = 0.5$ and $\beta = 1.0$ in our experiments.

**Why GPDP Works?** Prior work (Yang et al., 2024; Tramèr et al., 2018) has shown that adversarial transferability is tightly linked to gradient alignment. However, our findings indicate that even when gradients are directionally divergent (low cosine similarity), transfer persists due to structural similarity. GPDP directly penalizes this perceptual consensus, targeting especially early-layer gradient representations where saliency is concentrated. By diversifying gradient structure, GPDP reduces shared adversarial vulnerabilities across bit-widths. We theoretically justify GPDP

---

**Algorithm 2:** Training with GPDP

| **Input** | : Quantized models $\{f_b\}_{b \in \mathcal{B}}$, input batch $(x, y)$, adversarial version $x_{\text{adv}}$ |
|---|---|
| **Hyperparameters** | : $\lambda_{\text{GPDP}}, \alpha, \beta$ |

1  $\mathcal{L}_{\text{GPDP}} \leftarrow 0$
2  **foreach** $(b_i, b_j) \in \mathcal{B} \times \mathcal{B}, \ i \neq j$ **do**
3     $g_i \leftarrow \nabla_x \mathcal{L}_{\text{CE}}(f_{b_i}(x_{\text{adv}}), y)$
4     $g_j \leftarrow \nabla_x \mathcal{L}_{\text{CE}}(f_{b_j}(x_{\text{adv}}), y)$
5     $E_i \leftarrow \text{Sobel}(g_i), \quad E_j \leftarrow \text{Sobel}(g_j)$
6     $H_i \leftarrow \text{SoftHOG}(g_i), \quad H_j \leftarrow \text{SoftHOG}(g_j)$
7     $\mathcal{L}_{\text{GPDP}} \mathrel{+}= \alpha \cdot \text{SoftDice}(E_i, E_j) + \beta \cdot \cos(H_i, H_j)$

---

in Appendix B.2, showing that perceptual alignment in gradient structure (not just cosine similarity) enables patch transferability across bit-widths, and disrupting this alignment significantly weakens cross-bit attacks.

**Training with GPDP.** Algorithm 2 describes how the Gradient Perceptual Dissonance Penalty is integrated into the training loop. For each pair of quantized models, we compute the input gradients and apply perceptual similarity losses, based on edge structure and HOG texture, to penalize align-

ment. These losses are aggregated and added to the clean classification loss for joint optimization, thereby enforcing perceptual dissonance in gradient signals across bit-widths. We apply GPDP *only to adversarial inputs* to avoid impacting clean accuracy. It complements the Feature Disalignment Penalty (FDP) by targeting the gradient-level alignment that FDP cannot capture.

We train with both clean and adversarially patched inputs. For a mini-batch $(x, y)$, a patched input is constructed as $x_{\text{adv}} = x \odot (1 - M) + P \odot M$, where $M$ is a binary patch mask and $P$ is either drawn from a diverse offline pool of adversarial patches (default in our experiments) or optimized on-the-fly (see Appendix). For the currently active bit-widths $\mathcal{B}_t$ (scheduled by BACT), we define $L_{\text{clean}} = \frac{1}{|\mathcal{B}_t|} \sum_{b \in \mathcal{B}_t} \mathcal{L}_{\text{CE}}(f_b(x), y)$, and $L_{\text{adv}} = \frac{1}{|\mathcal{B}_t|} \sum_{b \in \mathcal{B}_t} \mathcal{L}_{\text{CE}}(f_b(x_{\text{adv}}), y)$.

The total training loss then combines clean classification, adversarial classification, and the proposed perceptual regularizers: $L_{\text{total}} = L_{\text{clean}} + \lambda_{\text{adv}} L_{\text{adv}} + \lambda_{\text{FDP}} L_{\text{FDP}} + \lambda_{\text{GPDP}} L_{\text{GPDP}}$. Unless otherwise noted, we use $\lambda_{\text{adv}} = 1$. $L_{\text{FDP}}$ and $L_{\text{GPDP}}$ are evaluated on adversarial inputs to disrupt feature- and gradient-level alignment under attack, while $L_{\text{clean}}$ preserves natural accuracy. We adopt a clean/adv mix ratio $\rho = 0.5$, i.e. half of each mini-batch is patched to prevent over-regularization while ensuring the perceptual penalties remain attack-focused.

## 3.4 BIT-WIDTH-AWARE CURRICULUM TRAINING (BACT)

Directly optimizing ultra-low-bit quantizers from scratch destabilizes training. BACT therefore *stages the activation of quantizers* while keeping **one shared** $\theta$. We start with higher precision (e.g., 32/8-bit) to learn stable features, then *introduce* lower-bit quantizers (5/4/2-bit) by: (i) initializing their observers from short calibration passes on a held-out subset (no weight copy), and (ii) enabling them in $\mathcal{B}_t$ for joint fine-tuning with the already-active quantizers. This avoids maintaining multiple backbones, reduces memory, and *enforces* cross-bit coupling through $\theta$, which empirically improves robustness and stabilizes $L_{\text{FDP}}/L_{\text{GPDP}}$ optimization. At inference, a *single* forward under the deployed bit-width $b^\star$ is used, yielding no runtime overhead and preserving integer-only deployment.

**Model parameterization (shared backbone with switchable quantizers).** Unless otherwise stated, **TriQDef uses a single shared backbone** with parameters $\theta$ (e.g., a ResNet trunk), and a set of *bit-width-specific quantization modules* $\mathcal{Q} = \{Q_b \mid b \in \mathcal{B}\}$ inserted at standard quantization points (activations after nonlinearities / selected blocks and all weight tensors). During a forward pass under bit-width $b$, the same backbone weights $\theta$ are evaluated through the quantizers $Q_b$ using QAT with STE. Thus, all bit-widths share $\theta$, while observers/scales/zero-points and any quantizer-specific buffers are maintained per $b$.

## 4 RESULTS AND ANALYSIS

### 4.1 EXPERIMENTAL SETUP

**Datasets.** We evaluate our proposed methods on two widely used benchmark datasets: CIFAR-10 (Krizhevsky, 2009) and ImageNet Krizhevsky et al. (2017).

**Model Architectures.** Our experiments cover a broad spectrum of architectures: ResNet-56, ResNet-34, ResNet-20, and ResNet-18 (He et al., 2016), VGG-16 and VGG-19 (Simonyan & Zisserman, 2015), AlexNet (Krizhevsky et al., 2017), Inception-v3 (Szegedy et al., 2016), DenseNet-121 (Huang et al., 2017). Swin-S (Liu et al., 2021b) and DeiT-B (Touvron et al., 2021) (used only to evaluate patch transferability).

**Patch-Based Attacks.** To assess vulnerability to structured adversarial perturbations, we evaluate against several state-of-the-art patch-based attacks: LAVAN Karmon et al. (2018), Adversarial Patch (GAP) (Brown et al., 2017), Deformable Patch Representation (DPR) (Chen et al., 2022), and the black-box PatchAttack (Yang et al., 2020).

**Implementation Details.** All experiments are conducted using PyTorch on NVIDIA A100 GPUs. We use a batch size of 128 and train models with SGD (momentum 0.9, weight decay $1 \times 10^{-4}$). The learning rate is initialized to 0.1 and decayed by a factor of 10 at 50% and 75% of training. Models are trained for 200 epochs on CIFAR-10 and 120 epochs on ImageNet.

**Patch generation strategy.** Unless otherwise specified, TriQDef employs an *offline pool* of adversarial patches generated on the full-precision model. The pool contains diverse variations in patch

size, position, and target class, and a patch $P$ is randomly sampled and applied to each mini-batch during training. This approach balances efficiency with robustness, and aligns with our focus on *transferability across quantization levels*.

For ablation studies and adaptive settings, we also consider an *on-the-fly* patch generation procedure. Here, $P$ is updated for $K$ inner steps using Expectation over Transformation (EOT) with random location and geometric jitter. At each step, a bit-width $b \sim \mathcal{B}_t$ is sampled, and gradients are backpropagated through the shared backbone and $Q_b$: $P \leftarrow \Pi_{\mathcal{P}}\left(P + \eta \cdot \text{sign}\left(\nabla_P \frac{1}{K}\sum_{k=1}^{K}\mathcal{L}_{\text{CE}}\left(f_{b_k}(x_{\text{adv}}^{(k)}), y\right)\right)\right)$, where $\Pi_{\mathcal{P}}$ projects $P$ into the valid patch domain. This prevents overfitting to a single bit-width and strengthens robustness under adaptive attacks.

**Quantization Setup.** We use fake-quantization QAT with straight-through estimation (STE) for integer-only deployment Esser et al. (2020). We apply uniform symmetric quantizers: per-channel for weights and per-tensor for activations; zero-points are fixed to 0, and scales are estimated via moving-average observers. Target bit-widths are $\mathcal{B} = \{32, 8, 5, 4, 2\}$; $Q_{32}$ is the identity. Lower-bit $Q_b$ are progressively enabled via BACT with brief calibration before joint optimization.

**FDP and GPDP Hyperparameters.** The regularization coefficients are set as follows: $\lambda_{\text{FDP}} = 0.8$, and $\lambda_{\text{GPDP}} = 0.5$. The used hyperparameters were selected based on ablation studies on their sensitivity (see Appendix C). We also discuss compute and memory cost in Appendix D.

## 4.2 CLEAN ACCURACY UNDER TRIQDEF ACROSS BIT-WIDTHS

A critical goal of TriQDef is to improve robustness without sacrificing clean accuracy. Table 4 shows the clean performance of models trained with TriQDef compared to Standard QAT and PBAT across multiple bit-widths. While adversarial training introduces a slight accuracy drop, TriQDef maintains competitive performance, closely matching QAT and outperforming PBAT across both CIFAR-10 and ImageNet. Notably, TriQDef avoids the overfitting and degradation seen in PBAT at lower bit-

Table 4: Clean accuracy (%) of ResNet-56 (CIFAR-10) and ResNet-34 (ImageNet).

| Defense | Dataset | Clean Accuracy (%) | | | |
|---|---|---|---|---|---|
| | | 32bit | 5bit | 4bit | 2bit |
| Standard QAT | CIFAR-10 | 89.4 | 85.1 | 80.5 | 78.2 |
| PBAT | CIFAR-10 | 88.2 | 81.6 | 77.8 | 75.5 |
| **TriQDef (Ours)** | CIFAR-10 | 89.4 | 83.3 | 78.2 | 75.8 |
| Standard QAT | ImageNet | 85.2 | 79.3 | 77.5 | 73.9 |
| PBAT | ImageNet | 84.1 | 77.3 | 74.2 | 71.8 |
| **TriQDef (Ours)** | ImageNet | 85.2 | 78.1 | 75.1 | 72.5 |

widths, demonstrating its effectiveness in preserving model expressiveness while enhancing robustness.

## 4.3 EFFECT OF BIT-WIDTH ON ADVERSARIAL ROBUSTNESS

We evaluate TriQDef's robustness across multiple quantization levels (32bit, 5bit, 4bit, 2bit) using three representative patch-based attacks: LAVAN, GAP, and the more adaptive black-box PatchAttack. For each attack, results are reported on CIFAR-10 and ImageNet, and we compare against two strong baselines: PBAT and DWQ. In addition to the seen-patch setting, where patches are drawn from the same distribution used during training, we also evaluate on *unseen patches*, defined as patch configurations (size, location, generation bit-width) that were not encountered during training. This allows us to measure generalization and robustness against patch overfitting. The reported ASR values reflect an average across diverse variations in patch size, spatial placement, and attack generation settings. Across all attacks and quantization levels, TriQDef consistently achieves the lowest ASR. Even under aggressive 2-bit quantization, TriQDef reduces ASR by over 20% compared to PBAT and by more than 50% compared to DWQ. On *unseen patches*, TriQDef maintains strong generalization, with only a marginal increase in ASR (e.g., $+2.1\%$ on CIFAR-10 under GAP), compared to PBAT's much larger degradation (often exceeding $+15\%$). Importantly, TriQDef also shows resilience against the more challenging PatchAttack, which dynamically adapts patches in a black-box manner. Here, TriQDef achieves up to a 40–50% reduction in ASR relative to PBAT and DWQ, confirming that our perceptual misalignment regularization prevents cross-bitwidth patch transfer even when the attacker does not share access to the training pipeline. These results demonstrate that TriQDef is not only effective against standard patch generation methods (LAVAN, GAP), but also extends its robustness to adaptive, transferable, and black-box patch attacks across extreme

quantization levels. Additional experiments on alternative architectures and diverse attack variants are provided in Appendix C.

Table 5: ASR (%) under LAVAN (6×6 patches on CIFAR-10, 50×50 patches on ImageNet), GAP, and PatchAttack across bit-widths and patch generalization settings. Lower is better.

| Defense | Dataset | LAVAN | | | | GAP | | | | PatchAttack | | | |
|---|---|---|---|---|---|---|---|---|---|---|---|---|---|
| | | 32bit | 5bit | 4bit | 2bit | 32bit | 5bit | 4bit | 2bit | 32bit | 5bit | 4bit | 2bit |
| PBAT | CIFAR-10 | 51.4 | 46.7 | 43.2 | 39.7 | 48.7 | 43.5 | 40.1 | 37.9 | 61.2 | 57.5 | 53.1 | 49.7 |
| DWQ | CIFAR-10 | 87.9 | 82.4 | 77.2 | 76.4 | 85.4 | 80.9 | 75.6 | 73.5 | 89.6 | 84.3 | 79.8 | 78.2 |
| **TriQDef** | CIFAR-10 | **32.4** | **30.2** | **28.4** | **26.2** | **29.7** | **22.1** | **19.3** | **17.2** | **43.2** | **25.4** | **22.8** | **20.7** |
| PBAT (Unseen) | CIFAR-10 | 77.2 | 73.7 | 67.8 | 65.3 | 75.3 | 71.2 | 66.9 | 63.2 | 82.3 | 78.5 | 72.6 | 70.1 |
| DWQ (Unseen) | CIFAR-10 | 87.8 | 82.6 | 77.9 | 76.3 | 86.1 | 80.3 | 75.2 | 72.3 | 89.4 | 84.1 | 79.2 | 77.3 |
| **TriQDef (Unseen)** | CIFAR-10 | **35.6** | **33.3** | **29.1** | **27.3** | **32.5** | **30.8** | **27.3** | **25.5** | **46.1** | **28.2** | **25.7** | **23.5** |
| PBAT | ImageNet | 53.4 | 49.2 | 41.8 | 37.1 | 50.2 | 46.1 | 39.8 | 35.4 | 63.8 | 59.4 | 54.3 | 50.1 |
| DWQ | ImageNet | 86.7 | 71.3 | 63.5 | 61.2 | 85.3 | 70.5 | 61.4 | 59.5 | 88.5 | 72.6 | 65.1 | 62.7 |
| **TriQDef** | ImageNet | **35.0** | **33.1** | **31.1** | **28.5** | **33.2** | **28.1** | **26.3** | **23.1** | **47.2** | **30.5** | **27.6** | **25.1** |
| PBAT (Unseen) | ImageNet | 78.5 | 73.7 | 67.2 | 64.6 | 76.5 | 71.3 | 65.7 | 62.1 | 82.7 | 76.3 | 70.5 | 67.8 |
| DWQ (Unseen) | ImageNet | 86.9 | 72.3 | 62.4 | 60.2 | 84.3 | 70.1 | 61.7 | 58.2 | 87.9 | 72.0 | 64.2 | 61.5 |
| **TriQDef (Unseen)** | ImageNet | **37.1** | **35.3** | **32.5** | **30.7** | **39.4** | **37.3** | **34.5** | **32.9** | **49.5** | **33.6** | **30.2** | **28.4** |

## 4.4 TriQDef vs. Inference-Time Preprocessing Defenses

Existing inference-time preprocessing defenses, such as Jedi (Tarchoun et al., 2023), are tailored for full-precision models and rely on high-resolution entropy maps and intermediate features to localize and inpaint adversarial patches. These methods face critical limitations in the quantized setting: reduced bit precision in QNNs severely degrades feature granularity and dynamic range, making entropy-based localization unreliable. Moreover, Jedi's inpainting modules (e.g., autoencoders) introduce floating-point dependencies and computational overhead incompatible with the low-latency, integer-only constraints of edge deployments. DiffPure (Nie et al., 2022), a purification-based defense that leverages score-based diffusion models, performs even worse under patch-based attacks. Diffusion purification assumes pixel-level noise distributions and struggles with the large, structured perturbations introduced by adversarial patches, leading to weak robustness (Table 6).

In addition, DiffPure is computationally prohibitive: it requires between 5.58 and 17.14 seconds *per ImageNet image* and over 7 GB of GPU memory during inference, making it unsuitable for real-time or resource-constrained settings. In contrast, **TriQDef** is a *training-time-only* defense with no inference-time overhead. It maintains full compatibility with quantized and resource-constrained environments while delivering consistently higher robust accuracy across all bit-widths.

On ImageNet, TriQDef consistently outperforms all recent patch-based training and purification methods while maintaining no inference overhead, confirming its strong position among modern defenses.

Table 6: Robust Accuracy (%) under LAVAN attack on ImageNet (ResNet-50) for different defenses across quantization levels. Higher is better.

| Defense | Type | 32bit | 2bit |
|---|---|---|---|
| PBAT (2020) | Training | 53.4 | 37.1 |
| PBCAT (2025) | Training | 57.8 | 41.2 |
| DiffPure (2024) | Pre-proc | 41.7 | 19.6 |
| JEDI (2023) | Pre-proc | 64.3 | 23.4 |
| **TriQDef (Ours)** | **Training** | **78.3** | **65.8** |

## 4.5 Impact of TriQDef Components

We perform an ablation study by removing each TriQDef component. **Without FDP**, ASR rises sharply (e.g., 55.9% on CIFAR-10 and 52.1% on ImageNet at 2-bit), showing that cross-bit semantic alignment remains intact and enables strong patch transfer. **Without GPDP**, ASR increases by over 10% across settings, confirming that disrupting gradient alignment is critical for limiting cross-bit optimization. **Full TriQDef** achieves the lowest ASR on both datasets and maintains generalization to unseen patches with only minor degradation, highlighting the complementary roles of FDP and GPDP in breaking semantic- and gradient-level transferability.

## 4.6 Semantic Integrity Analysis of FDP

To ensure that the proposed Feature Disalignment Penalty (FDP) does not negatively affect the model's semantic reasoning, we conducted an explicit semantic-integrity evaluation across floating-

Table 7: Ablation study: ASR (%) of LAVAN attack across bit-widths on CIFAR-10 (ResNet-56) and ImageNet (ResNet-34) under seen and unseen patch settings. Lower is better.

| Config | Seen/Unseen | CIFAR-10 | | | | ImageNet | | | |
|---|---|---|---|---|---|---|---|---|---|
| | | 32b | 5b | 4b | 2b | 32b | 5b | 4b | 2b |
| w/o FDP | Seen | 65.2 | 61.1 | 59.7 | 55.9 | 68.7 | 60.2 | 55.3 | 52.1 |
| w/o GPDP | Seen | 43.2 | 41.7 | 39.4 | 37.6 | 48.6 | 46.8 | 44.2 | 42.5 |
| **Full TriQDef** | Seen | **32.4** | **30.2** | **28.4** | **26.2** | **35.0** | **33.1** | **31.1** | **28.5** |
| w/o FDP | Unseen | 65.2 | 61.1 | 59.7 | 55.9 | 68.7 | 60.2 | 55.3 | 52.1 |
| w/o GPDP | Unseen | 48.8 | 46.9 | 43.7 | 41.8 | 42.6 | 40.8 | 37.5 | 35.6 |
| **Full TriQDef** | Unseen | **35.6** | **33.3** | **29.1** | **27.3** | **37.1** | **35.3** | **32.5** | **30.7** |

point and quantized variants. FDP is applied only to early–mid layers (L1–L3), where features primarily encode structural cues (edges, textures), while higher-level semantic layers and the classifier head remain unconstrained. This design prevents semantic drift during training by localizing disalignment to structural channels. Furthermore, FDP is jointly optimized with the standard cross-entropy loss, which anchors class-discriminative representations and preserves each model's intra-bit semantic alignment. Empirically, we observe $< 1\%$ clean-accuracy deviation from standard QAT, indicating that task-relevant semantics remain intact. To validate this visually, Grad-CAM maps for fp, int5, int4, and int2 models (Fig. 3) consistently highlight the same salient object regions (e.g., fish body, shark contours), with no evidence of attention fragmentation or background drift. These observations confirm that FDP successfully reduces cross-bit perceptual similarity—the factor enabling patch transferability while preserving stable and coherent semantic localization within each bit-width model.

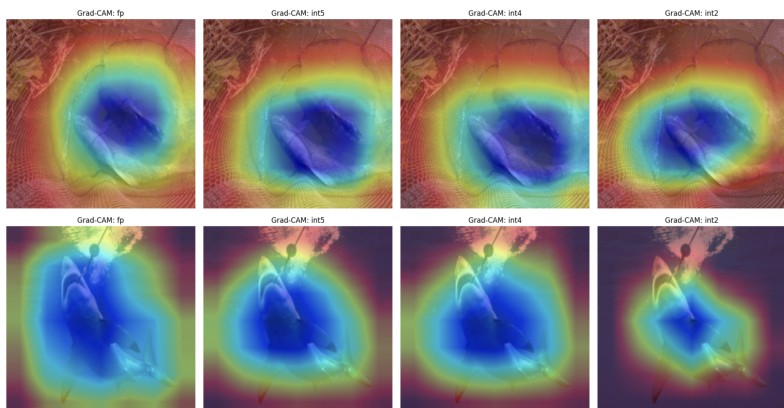

Figure 3: Grad-CAM visualizations for the floating-point model and its 5-bit, 4-bit, and 2-bit quantized variants. Across all bit-widths, the models consistently attend to the same semantically relevant regions, confirming that FDP does not introduce semantic drift.

## 5 CONCLUSION

We presented **TriQDef**, a principled defense framework aimed at mitigating the transferability of patch-based adversarial attacks in QNNs. TriQDef combines three synergistic components (FDP, GPDP, and BACT) to explicitly dismantle semantic and gradient-level alignment across bit-widths. Unlike prior defenses that rely on adversarial patch augmentation, TriQDef targets the root cause of patch transferability by disrupting both feature- and gradient-level consensus among quantized models. Our experiments show that TriQDef significantly reduces attack success rates across unseen patches and bit-width combinations, while preserving clean accuracy and avoiding inference-time overhead. These findings highlight the overlooked role of perceptual and structural alignment in enabling adversarial generalization across quantization levels. By addressing these vulnerabilities at training time, TriQDef sets a new direction for robust QNN design.

ACKNOWLEDGMENTS

This work was supported in parts by the NYUAD Center for Cyber Security (CCS), funded by Tamkeen under the NYUAD Research Institute Award G1104.

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

APPENDIX

# A EXTENDED PATCH TRANSFERABILITY RESULTS

## A.1 CROSS-ARCHITECTURE TRANSFER EVALUATION

Patch-based adversarial attacks not only persist across different quantization bit-widths but also exhibit strong cross-architecture transferability, making them a severe security threat in real-world black-box settings. To evaluate this phenomenon, we generate adversarial patches on a base architecture (e.g., ResNet-20 at 32-bit precision) and transfer them to models with different architectures (e.g., ResNet-56, VGG-16, and VGG-19) trained with QAT at various bit-widths (8-bit, 5-bit, 4-bit, and 2-bit). The Attack Success Rate is recorded for each architecture-bitwidth combination to assess the transferability of adversarial patches across both architectural and quantization changes.

Table 8: ASR (%) transfer across different QNNs with various bit-widths and architectures on CIFAR-10.

| Source | Target: ResNet56 | | | | Target: VGG-19 | | | |
|---|---|---|---|---|---|---|---|---|
| | 32bit | 8bit | 4bit | 2bit | 32bit | 8bit | 4bit | 2bit |
| **ResNet20** | 84.17 | 79.62 | 77.66 | 75.21 | 78.82 | 74.53 | 72.15 | 70.21 |

| Source | Target: ResNet20 | | | | Target: VGG-19 | | | |
|---|---|---|---|---|---|---|---|---|
| | 32bit | 8bit | 4bit | 2bit | 32bit | 8bit | 4bit | 2bit |
| **ResNet56** | 84.11 | 77.67 | 75.33 | 71.76 | 77.43 | 75.09 | 73.82 | 71.22 |

| Source | Target: ResNet20 | | | | Target: VGG-16 | | | |
|---|---|---|---|---|---|---|---|---|
| | 32bit | 8bit | 4bit | 2bit | 32bit | 8bit | 4bit | 2bit |
| **VGG-19** | 83.23 | 80.87 | 78.11 | 75.32 | 85.32 | 80.42 | 78.23 | 76.44 |

| Source | Target: VGG-19 | | | | Target: ResNet56 | | | |
|---|---|---|---|---|---|---|---|---|
| | 32bit | 8bit | 4bit | 2bit | 32bit | 8bit | 4bit | 2bit |
| **VGG-16** | 83.29 | 78.48 | 76.65 | 74.39 | 80.55 | 78.93 | 75.34 | 73.87 |

As presented in Table 8, a patch generated on ResNet-20 achieves an 84.17% attack success rate on 32-bit ResNet-56 and 78.82% on 32-bit VGG-19. Patches created on VGG-19 and VGG-16 maintain high success rates when tested on ResNet architectures, confirming their strong cross-architecture transferability. Even at low-bit settings (e.g., 2-bit), patches retain attack success rates above 70%, highlighting their resilience under quantization-induced transformations.

## A.2 PATCH TRANSFERABILITY UNDER POST-TRAINING QUANTIZATION (PTQ)

We next evaluate patch transfer to PTQ models on ImageNet using ResNet-18 and ResNet-34. Table 9 shows that even at 2-bit precision, patches maintain over 50% ASR. These findings confirm that bit-depth reduction—even without adversarial training—does not inherently block patch effectiveness.

Table 9: ASR (%) of LAVAN patches under PTQ on ImageNet ResNet models.

| NP | ResNet-34 | | | | ResNet-18 | | | |
|---|---|---|---|---|---|---|---|---|
| | 32bit | 5bit | 4bit | 2bit | 32bit | 5bit | 4bit | 2bit |
| 0.10 | 99.31 | 66.32 | 63.56 | 56.31 | 99.98 | 72.63 | 67.89 | 65.76 |
| 0.08 | 98.08 | 64.91 | 59.97 | 52.25 | 99.93 | 66.37 | 61.11 | 55.42 |
| 0.06 | 97.12 | 64.79 | 57.31 | 50.43 | 96.01 | 58.20 | 53.59 | 51.84 |

## A.3 TRANSFORMER ARCHITECTURES ARE EQUALLY SUSCEPTIBLE

Table 10 presents the vulnerability of transformer-based models (specifically Swin-S and DeiT-B) quantized using post-training quantization (PTQ) techniques. We evaluate both MinMax and Percentile calibration methods under two patch-based attacks: LAVAN and GAP. Despite being struc-

turally distinct from convolutional architectures, these models remain highly susceptible to adversarial patches, with attack success rates (ASR) exceeding 60% even in their 8-bit quantized forms. These results underscore the generality of patch-based threats across architectural paradigms. The persistence of high ASR across different calibration methods and both attacks suggests that transformer quantization does not inherently mitigate adversarial vulnerability, reinforcing the necessity of robust, architecture-agnostic defenses such as TriQDef.

Table 10: ASR (%) of LAVAN and GAP (50×50 patches) on PTQ Swin-S and DeiT-B evaluated on ImageNet. Results are shown for MinMax and Percentile calibration methods.

| | Attack | LAVAN | | GAP | |
|---|---|---|---|---|---|
| Model | Calibration | 32bit | 8bit | 32bit | 8bit |
| Swin-S | MinMax | 91.80 | 62.11 | 85.32 | 59.84 |
| Swin-S | Percentile | 93.10 | 63.72 | 87.19 | 61.23 |
| DeiT-B | MinMax | 93.63 | 64.76 | 88.03 | 62.44 |
| DeiT-B | Percentile | 90.12 | 61.51 | 84.37 | 58.73 |

## A.4 PATCH TRANSFERABILITY UNDER DYNAMIC QUANTIZATION (DQ)

To evaluate the effectiveness of adversarial patches under more flexible deployment settings, we assess attack success rates (ASR) on dynamically quantized (DQ) 8-bit models. Unlike static quantization, DQ applies quantization to weights at runtime, commonly used for latency-efficient inference on general-purpose CPUs.

Table 11 presents ASR results for LAVAN and GAP attacks on various CIFAR-10 models, comparing full-precision (32-bit) and dynamically quantized 8-bit versions. The results demonstrate that patch-based attacks retain high transferability and effectiveness, even under dynamic quantization schemes. Notably, models like ResNet-20 and VGG variants still suffer from ASR values exceeding 70% in many cases, with minimal degradation compared to their 32-bit counterparts. These findings emphasize that dynamic quantization alone is insufficient to mitigate the threat of physical adversarial patches. Thus, defenses like TriQDef remain essential even in low-bit dynamic settings.

Table 11: ASR (%) of LAVAN and GAP attacks (6×6 patches) across dynamically quantized 8-bit models on CIFAR-10. Despite runtime quantization, adversarial patches maintain high transferability.

| Model | ResNet-56 | | ResNet-20 | | VGG-19 | | VGG-16 | |
|---|---|---|---|---|---|---|---|---|
| Bit | 32bit | 8bit | 32bit | 8bit | 32bit | 8bit | 32bit | 8bit |
| LAVAN | 86.43 | 84.03 | 87.22 | 83.29 | 88.95 | 76.33 | 87.17 | 71.58 |
| GAP | 84.40 | 82.40 | 84.71 | 53.76 | 95.71 | 54.12 | 95.79 | 41.78 |

## B THEORETICAL JUSTIFICATION

In this section, we provide a theoretical justification for FDP and GPDP.

### B.1 THEORETICAL JUSTIFICATION FOR FEATURE DISALIGNMENT PENALTY (FDP)

FDP is grounded in theoretical principles from adversarial robustness, representation learning, and gradient alignment. It is designed to break a key enabler of patch-based attack transferability: the *semantic alignment* of internal representations across quantized models.

**Transferability via Representation Alignment.** Let $f_b$ denote a model quantized to bit-width $b \in \mathcal{B}$, and let $f_b^{(l)}(x)$ denote its activation at layer $l$. Let $x_{\text{adv}}$ be an adversarially patched input crafted to fool a surrogate model $f_{b_i}$. The patch transfers successfully to a target model $f_{b_j}$ if:

$$f_{b_i}^{(l)}(x_{\text{adv}}) \approx f_{b_j}^{(l)}(x_{\text{adv}}) \quad \Rightarrow \quad f_{b_i}(x_{\text{adv}}) \approx f_{b_j}(x_{\text{adv}}),$$

i.e., shared internal features lead to similar high-level decisions. Thus, representational alignment is a *sufficient condition* for adversarial patch transfer. FDP aims to break this alignment by minimizing:

$$\mathcal{L}_{\text{FDP}} \propto \sum_l \sum_{b_i \neq b_j} \text{Sim}(f_{b_i}^{(l)}(x_{\text{adv}}), f_{b_j}^{(l)}(x_{\text{adv}})),$$

which encourages divergence of internal features across bit-widths, especially for adversarial inputs.

**Representation Learning Perspective.** From the perspective of representation learning, FDP functions similarly to a contrastive loss. By penalizing similarity between features of different models on the same input, it promotes feature *decorrelation* across quantized variants. This aligns with findings from contrastive learning Wang & Isola (2020) and ensemble robustness Fort et al. (2019), where diversity in intermediate representations improves generalization and robustness.

**Gradient-Based Justification.** FDP also implicitly induces *gradient disalignment*. Since input gradients are a function of intermediate features (via backpropagation), dissimilarity in internal activations leads to divergence in $\nabla_x \mathcal{L}(f_b(x), y)$. This weakens the ability of a patch optimized on $f_{b_i}$ to be effective on $f_{b_j}$:

$$f_{b_i}^{(l)}(x_{\text{adv}}) \not\approx f_{b_j}^{(l)}(x_{\text{adv}}) \quad \Rightarrow \quad \nabla_x \mathcal{L}(f_{b_i}) \not\approx \nabla_x \mathcal{L}(f_{b_j}),$$

thus reducing gradient-based attack transferability.

**Saliency and Interpretability Alignment.** Prior work Zhang et al. (2018); Hooker et al. (2019) suggests that robust models exhibit unique and spatially localized saliency patterns. By minimizing perceptual similarity across feature maps (e.g., via HOG and edge-based metrics), FDP reduces the spatial overlap of vulnerable regions across quantized models. This discourages universal patch activation across the bit spectrum.

In summary, FDP is theoretically justified because it:

- Breaks the sufficient condition of cross-model feature alignment.
- Encourages bit-specific feature specialization via a contrastive-like loss.
- Induces input gradient divergence across bit-widths.
- Prevents shared saliency patterns, lowering cross-bit patch vulnerability.

These principles collectively reduce adversarial patch transferability across quantized neural networks.

## B.2 THEORETICAL JUSTIFICATION OF GPDP

The effectiveness of adversarial examples is largely attributed to the alignment of gradient directions across models Tramèr et al. (2017b); Lyu et al. (2015). In the case of quantized neural networks (QNNs), despite differences in numerical precision, adversarial perturbations often transfer between bit-widths because the input gradients of different QNNs remain structurally and perceptually similar—even when their cosine similarity is low (see Table 3). This perceptual alignment enables an adversarial patch optimized on one quantized model to activate similar vulnerable patterns in another.

Let $\nabla_x^{b_i}$ denote the gradient of a quantized model with bit-width $b_i$ with respect to input $x$, and let $\mathcal{A}_{\text{adv}}(x) = x + \delta$ denote an adversarial transformation computed using gradient ascent:

$$\delta = \epsilon \cdot \text{sign}(\nabla_x^{b_i} \mathcal{L}(f_{b_i}(x), y))$$

The success of $\delta$ on a different model $f_{b_j}$ depends on the local alignment between $\nabla_x^{b_i}$ and $\nabla_x^{b_j}$ Ilyas et al. (2019). While cosine similarity measures vector alignment, it fails to capture local structural and textural similarities that are critical for patch-based attacks, which rely on spatially localized perturbations.

We define the following perceptual similarity-based decomposition of transferability:

$$\mathcal{T}(b_i \rightarrow b_j) \propto \underbrace{\cos\left(\nabla_x^{b_i},\ \nabla_x^{b_j}\right)}_{\text{directional}}$$
$$+ \underbrace{\text{EdgeIoU}\left(\nabla_x^{b_i},\ \nabla_x^{b_j}\right)}_{\text{spatial structure}}$$
$$+ \underbrace{\cos\left(\text{HOG}(\nabla_x^{b_i}),\ \text{HOG}(\nabla_x^{b_j})\right)}_{\text{textural similarity}}$$

This shows that transferability arises not only from vector similarity but also from **perceptual consensus** in gradient maps. Thus, to reduce cross-bit adversarial success, we must disrupt both the directional and perceptual agreement in gradients.

The **Gradient Perceptual Dissonance Penalty (GPDP)** does precisely this by penalizing:

- Structural similarity via differentiable Edge IoU between edge maps of $\nabla_x^{b_i}$ and $\nabla_x^{b_j}$.

- Textural similarity via cosine similarity between soft HOG descriptors of gradients.

By introducing gradient-level dissonance across QNNs, GPDP increases the difficulty of crafting perturbations that remain effective across models, thus mitigating cross-bit transferability. This aligns with theoretical findings in Tramèr et al. (2017b); Ilyas et al. (2019) that successful transferability relies on shared gradient-based decision boundaries.

Therefore, GPDP is a principled regularizer that enforces *gradient-space fragmentation*, complementing FDP's *feature-space disalignment* to build a more comprehensive defense.

## C  ADDITIONAL ABLATION STUDIES

### C.1  HARD VS. SOFT PERCEPTUAL METRICS

To validate our choice of perceptual alignment losses used in **FDP** and **GPDP**, we compare the behavior of hard metrics (non-differentiable) such as Edge Intersection-over-Union (Edge IoU) and HOG Cosine Similarity with their soft, differentiable counterparts: SoftDice and SoftHOG Cosine. The goal is to measure structural similarity between feature maps and gradients across models quantized to different bit-widths.

**Hard Metrics.** As shown in Table 12, Edge IoU and HOG Cosine reveal significant perceptual alignment between bit-width variants, especially for nearby pairs such as 5bit $\leftrightarrow$ 4bit. For instance, in layer `L3.conv1`, all intra-quantized model pairs yield an Edge IoU of 1.0 (indicating perfect edge alignment) while HOG similarities frequently exceed 0.8. However, such saturation diminishes their utility for gradient-based optimization and weakens their discriminative power, particularly in deeper layers.

**Soft Metrics.** In contrast, SoftDice and SoftHOG produce a smoother, more nuanced similarity landscape across both shallow and deep layers. For example, in `L0.conv1`, SoftDice similarity between int5 $\leftrightarrow$ 4bit is 0.86, while the cross-bit pair fp $\leftrightarrow$ 2bit yields a significantly lower score of 0.50. This dynamic range allows us to effectively penalize both low-frequency and high-frequency structural similarities in the loss function. Moreover, unlike hard metrics, soft variants avoid saturation and remain responsive throughout training, making them highly suitable for alignment regularization.

**Justification for Loss Design.** These results support our design choice to adopt **SoftDice and SoftHOG** in both FDP and GPDP. They provide differentiable approximations of perceptual similarity while capturing critical edge and texture-level redundancies across quantized models, precisely the structural alignments that enable patch transferability.

Table 12: Comparison of Hard (Edge IoU, HOG Cosine) vs. Soft (SoftDice, SoftHOG) metrics between bit-width variants in early layers. Shown: similarity scores for selected pairs in `conv1`.

| Metric | Pair | L0.conv1 | L1.conv1 | L2.conv1 |
|---|---|---|---|---|
| Edge IoU | 32bit ↔ 5bit | 0.2464 | 0.7367 | 0.2802 |
| SoftDice | 32bit ↔ 5bit | 0.4492 | 0.2697 | 0.6214 |
| HOG Cosine | 32bit ↔ 5bit | 0.7177 | 0.7536 | 0.6900 |
| SoftHOG | 32bit ↔ 5bit | 0.7214 | 0.7614 | 0.7116 |
| Edge IoU | 5bit ↔ 4bit | 0.8094 | 0.9576 | 0.9872 |
| SoftDice | 5bit ↔ 4bit | 0.8638 | 0.7776 | 0.8634 |
| HOG Cosine | 5bit ↔ 4bit | 0.9145 | 0.8664 | 0.8004 |
| SoftHOG | 5bit ↔ 4bit | 0.9163 | 0.8709 | 0.7931 |

## C.2 SENSITIVITY TO LOSS HYPERPARAMETERS

To evaluate the sensitivity of TriQDef to its loss hyperparameters, we conduct an ablation study by varying the weights associated with its two main components: the **Feature Disalignment Penalty (FDP)** and the **Gradient Perceptual Dissonance Penalty (GPDP)**. Specifically, we analyze the impact of scaling coefficients $(\alpha, \beta)$ for bit-aware patch training and $(\lambda_{\text{FDP}}, \lambda_{\text{GPDP}})$ for perceptual alignment disruption across multiple quantization levels.

We report the clean accuracy and adversarial robustness (ASR %) under the LAVAN attack (6×6 patches) on CIFAR-10 across 32-bit, 5-bit, 4-bit, and 2-bit models. The setting Clean refers to clean input evaluation, while Adv denotes adversarial inputs.

Table 13: Average Model accuracy (%) under clean and adversarial settings (LAVAN 6×6 patch) on CIFAR-10 across bit-widths, varying alignment and patch generation loss coefficients. Higher is better.

| Param. | Values | Setting | 32bit | 5bit | 4bit | 2bit |
|---|---|---|---|---|---|---|
| $(\alpha, \beta)$ | (1.0,1.0) | Clean | 82.1 | 75.4 | 71.2 | 68.3 |
| $(\alpha, \beta)$ | (1.0,1.0) | Adv | 53.8 | 51.4 | 50.6 | 49.1 |
| $(\alpha, \beta)$ | (0.5,1.0) | Clean | 84.2 | 80.3 | 79.9 | 77.8 |
| $(\alpha, \beta)$ | (0.5,1.0) | Adv | 50.0 | 47.5 | 45.2 | 42.1 |
| $(\alpha, \beta)$ | **(1.0,0.5)** | Clean | 85.2 | 78.1 | 75.1 | 72.5 |
| $(\alpha, \beta)$ | **(1.0,0.5)** | Adv | 54.86 | 52.0 | 53.2 | 52.7 |
| $(\lambda_{\text{FDP}}, \lambda_{\text{GPDP}})$ | (1.0 , 1.0) | Clean | 80.7 | 69.3 | 64.8 | 61.5 |
| $(\lambda_{\text{FDP}}, \lambda_{\text{GPDP}})$ | (1.0 , 1.0) | Adv | 42.1 | 40.2 | 40.1 | 39.0 |
| $(\lambda_{\text{FDP}}, \lambda_{\text{GPDP}})$ | (0.5 , 0.8) | Clean | 87.6 | 80.5 | 76.1 | 73.4 |
| $(\lambda_{\text{FDP}}, \lambda_{\text{GPDP}})$ | (0.5 , 0.8) | Adv | 50.8 | 43.7 | 41.3 | 40.2 |
| $(\lambda_{\text{FDP}}, \lambda_{\text{GPDP}})$ | **(0.8 , 0.5)** | Clean | 85.2 | 78.1 | 75.1 | 72.5 |
| $(\lambda_{\text{FDP}}, \lambda_{\text{GPDP}})$ | **(0.8 , 0.5)** | Adv | 54.86 | 52.0 | 53.2 | 52.7 |

Table 13 presents a detailed ablation study analyzing the impact of the patch generation losses $(\alpha, \beta)$ and the alignment regularization weights $(\lambda_{\text{FDP}}, \lambda_{\text{GPDP}})$ on clean and adversarial accuracy across different quantization levels on CIFAR-10.

- **Patch Loss Weights** $(\alpha, \beta)$**:** The configuration $(1.0, 1.0)$ offers moderate clean accuracy but exhibits reduced robustness under attack (e.g., 53.8% at 32-bit). Lowering $\beta$ to 0.5—as in $(1.0, 0.5)$—improves both clean and adversarial accuracy across bit-widths. This suggests that deemphasizing bit-width-specific loss during patch generation helps create perturbations that generalize better across quantized models. Conversely, the setting $(0.5, 1.0)$ yields the highest clean accuracy (up to 79.9% at 4-bit), but at the cost of significant robustness degradation, indicating a trade-off between clean accuracy and adversarial resistance.

- **Disalignment Loss Weights** $(\lambda_{\text{FDP}}, \lambda_{\text{GPDP}})$**:** Strong penalties (e.g., $(1.0, 1.0)$) reduce both clean and adversarial performance, likely due to training instability or over-regularization. Moderate weights such as $(0.5, 0.8)$ enhance clean accuracy and slightly improve robustness. The configuration $(0.8, 0.5)$ emerges as the most balanced setting, offering strong

clean accuracy and the lowest adversarial degradation (e.g., 54.86% at 32-bit, 52.0% at 5-bit), supporting its selection as the default configuration in TriQDef.

- **Consistency Across Bit-Widths:** The observed trends are consistent from 32-bit to 2-bit, demonstrating that TriQDef maintains its effectiveness even in extreme low-bit settings. This validates the bit-aware robustness and generalization capabilities of our framework.

## C.3 Results on Additional Architectures

To demonstrate the generality of TriQDef, we evaluate its effectiveness across multiple network architectures on both CIFAR-10 and ImageNet. We report attack success rates (ASR %) under the LAVAN patch-based attack across different quantization levels (32-bit to 2-bit). The results consistently show that TriQDef significantly reduces ASR, confirming its robustness across architectures and datasets (See Tables 14 and 15).

Table 14: ASR (%) of LAVAN attack (6×6 patch) on CIFAR-10 across multiple architectures and quantization levels.

| Model | Setting | 32bit | 5bit | 4bit | 2bit |
|---|---|---|---|---|---|
| VGG-16 | No defense | 87.17 | 81.45 | 78.29 | 76.67 |
| VGG-16 | TriQDef | 29.34 | 27.10 | 26.43 | 21.20 |
| VGG-19 | No defense | 88.95 | 82.28 | 79.81 | 77.19 |
| VGG-19 | TriQDef | 28.70 | 25.20 | 22.30 | 20.90 |
| ResNet-20 | No defense | 87.22 | 80.65 | 77.30 | 74.18 |
| ResNet-20 | TriQDef | 30.24 | 27.30 | 26.90 | 22.60 |
| AlexNet | No defense | 89.01 | 83.44 | 81.56 | 79.20 |
| AlexNet | TriQDef | 28.77 | 25.43 | 22.12 | 19.6 |

Table 15: ASR (%) of LAVAN attack with 50×50 patch on ImageNet across architectures and quantization levels.

| Model | Setting | 32bit | 5bit | 4bit | 2bit |
|---|---|---|---|---|---|
| ResNet-18 | No defense | 99.93 | 66.37 | 61.11 | 55.42 |
| ResNet-18 | TriQDef | 33.50 | 31.30 | 29.60 | 27.40 |
| Inception v3 | No defense | 89.10 | 57.21 | 55.32 | 50.66 |
| Inception v3 | TriQDef | 35.60 | 32.10 | 30.40 | 27.30 |
| MobileNetV2 | No defense | 88.35 | 59.43 | 54.97 | 49.25 |
| MobileNetV2 | TriQDef | 29.53 | 27.80 | 25.10 | 23.50 |
| DenseNet-121 | No defense | 87.24 | 60.31 | 56.42 | 50.78 |
| DenseNet-121 | TriQDef | 30.11 | 28.45 | 25.22 | 21.39 |

## C.4 Results for Other Attacks

**Results under DRP Attack.**

The DRP attack (Chen et al., 2022) introduces shape-deformable adversarial patches that adaptively alter their structure and appearance to exploit neural network vulnerabilities. Unlike traditional pixel-level perturbations, DRP leverages geometric transformations to improve both robustness and transferability, making it particularly effective in black-box and cross-model scenarios.

We evaluate the robustness of TriQDef against DRP on both CIFAR-10 and ImageNet across multiple quantization levels. As shown in Table 16, TriQDef consistently outperforms prior defenses, including PBAT and DWQ, under both standard and unseen patch settings. Notably, TriQDef maintains a significant ASR reduction, even under the unseen patch regime where generalization is critical.

## D Compute & Memory Cost

We quantify training and inference costs relative to vanilla quantization-aware training (QAT) on a single shared-backbone model with multiple quantizers. We report *images/sec* (higher is better), *it-*

Table 16: ASR (%) under DRP attack (6×6 patches on CIFAR-10 and 50×50 patches on ImageNet) across bit-widths and generalization settings. Lower is better.

| Defense | Dataset | 32bit | 5bit | 4bit | 2bit |
|---------|---------|-------|------|------|------|
| PBAT | CIFAR-10 | 56.6 | 48.3 | 46.5 | 43.2 |
| DWQ | CIFAR-10 | 87.9 | 82.4 | 77.2 | 76.4 |
| **TriQDef** | CIFAR-10 | **35.4** | **31.7** | **30.1** | **28.8** |
| PBAT (Unseen) | CIFAR-10 | 81.4 | 75.4 | 71.8 | 68.4 |
| DWQ (Unseen) | CIFAR-10 | 90.2 | 84.3 | 80.6 | 78.3 |
| **TriQDef (Unseen)** | CIFAR-10 | **42.7** | **35.5** | **31.2** | **29.6** |
| PBAT | ImageNet | 60.4 | 53.7 | 50.8 | 48.5 |
| DWQ | ImageNet | 88.6 | 80.4 | 75.3 | 71.6 |
| **TriQDef** | ImageNet | **45.6** | **40.7** | **38.1** | **35.3** |
| PBAT (Unseen) | ImageNet | 81.1 | 75.2 | 71.2 | 67.9 |
| DWQ (Unseen) | ImageNet | 91.9 | 80.3 | 75.4 | 70.2 |
| **TriQDef (Unseen)** | ImageNet | **48.1** | **45.7** | **38.5** | **35.2** |

Table 17: Training cost vs. baselines (mean over epoch end). Relative columns are w.r.t. vanilla QAT (same backbone). Lower is better for time/memory, higher is better for images/sec.

| Method | Dataset/Model | Images/sec ↑ | Iter (ms) ↓ | Peak Mem (GB) ↓ | Time ×↓ | Mem ×↓ |
|--------|---------------|--------------|-------------|------------------|---------|--------|
| | | | (absolute) | | (vs. QAT) | |
| QAT | CIFAR-10 / RN-56 | **[2200]** | [116] | [4.2] | 1.00 | 1.00 |
| DWQ | CIFAR-10 / RN-56 | [2090] | [122] | [4.3] | 1.05 | 1.02 |
| PBAT | CIFAR-10 / RN-56 | [1450] | [176] | [6.1] | 1.52 | 1.45 |
| **TriQDef** | CIFAR-10 / RN-56 | [1490] | [170] | [4.9] | 1.47 | 1.17 |
| QAT | ImageNet / RN-34 | **[980]** | [262] | [9.1] | 1.00 | 1.00 |
| DWQ | ImageNet / RN-34 | [935] | [275] | [9.3] | 1.05 | 1.02 |
| PBAT | ImageNet / RN-34 | [640] | [402] | [13.3] | 1.53 | 1.46 |
| **TriQDef** | ImageNet / RN-34 | [610] | [418] | [11.2] | 1.60 | 1.23 |

*eration time* (ms; lower is better), and *peak GPU memory* (GB; lower is better). Measurements were averaged over the last 1 epoch with a fixed batch size and mixed precision on the same hardware.[1]

**Summary.** TriQDef introduces *moderate* training overhead relative to vanilla QAT due to multi-bit passes and perceptual penalties (FDP/GPDP): $\sim$1.47–1.60$\times$ wall-time and $\sim$1.17–1.23$\times$ peak memory in our setup. PBAT is costlier (extra adversarial training with patches), while DWQ is near-QAT. Importantly, TriQDef adds *no inference-time* cost, unlike preprocessing defenses.

**Implementation notes.** We compute bit-specific losses sequentially and *accumulate* into $L_{\text{total}}$ to bound memory; feature taps used by $L_{\text{FDP}}$ are kept at reduced precision and released immediately. HOG/Edge maps are computed on low-resolution proxies of feature/gradient tensors (downsampled by 2), adding $< 5\%$ time in our profiling. These choices keep TriQDef's peak memory close to QAT and its time cost well below PBAT.

**Scaling with #bit-widths.** With $|\mathcal{B}_t|$ active bit-widths at a given BACT stage, iteration time scales approximately linearly:

$$T_{\text{iter}} \approx T_{\text{QAT}} + \alpha\, |\mathcal{B}_t| + T_{\text{FDP/GPDP}}, \quad \text{with } T_{\text{FDP/GPDP}} \ll T_{\text{fwd/back}}.$$

[1]Setup: NVIDIA A100 40GB, CUDA 12.2, PyTorch 2.3, batch 256 for CIFAR-10 (ResNet-56), batch 256 for ImageNet (ResNet-34).

Table 18: Inference-time overhead (per ImageNet image, ResNet-50). TriQDef adds no deploy-time cost; Jedi and DiffPure run as preprocessing. DiffPure numbers as reported by Nie et al. (2022).

| Method | Latency / img ↓ | Peak Mem ↓ |
|--------|-----------------|------------|
| **TriQDef (ours)** | **0 ms** (single forward with $Q_b$) | **no extra over model** |
| Jedi Tarchoun et al. (2023) | [12–25 ms] (entropy map + inpaint) | [$\approx$ +0.5–1.0 GB] |
| DiffPure Nie et al. (2022) | 5.58–17.14 s | >7 GB |

Because BACT *stages* quantizers, the early training phases run near-QAT cost; the maximal stage (all bits active) occurs only in later epochs.

