# OpenReview forum: "TriQDef: Disrupting Semantic and Gradient Alignment to Prevent Adversarial Patch Transferability in Quantized Neural Networks"
_ICLR.cc/2026/Conference — ICLR 2026 Poster_

### Official Review · Reviewer_V6ix · 2025-10-28

**Soundness:** 3
**Presentation:** 3
**Contribution:** 3
**Rating:** 8
**Confidence:** 3

**Summary:**

The paper introduces TriQDef, a tri-level quantization-aware defense framework designed to improve the robustness of Quantized Neural Networks (QNNs) against patch-based adversarial attacks. While quantization naturally distorts gradients and reduces vulnerability to pixel-level attacks, it remains weak against localized, transferable patch attacks across different bit-widths.

TriQDef mitigates this by integrating three components:
1. Feature Disalignment Penalty (FDP): penalizes perceptual similarity in intermediate features to enforce semantic inconsistency;
2. Gradient Perceptual Dissonance Penalty (GPDP): misaligns input gradients across quantization levels using structural metrics like Edge IoU and HOG cosine similarity;
3. Joint Quantization-Aware Training: optimizes these penalties jointly across multiple quantizers in a shared backbone.

Experiments on CIFAR-10 and ImageNet show that TriQDef reduces attack success rates by over 40% on unseen patch–quantization combinations while maintaining strong clean accuracy, demonstrating that disrupting both semantic and perceptual gradient alignment is key to improving QNN robustness.

**Strengths:**

1. The paper motivates the proposed components through quantitative analysis across different patch-based adversarial attack methods, QNN architectures, and numbers of quantized bits. With the quantitative results, the proposed components are easily convincing.

2. The proposed FDP and GPDP components are effectively breaking semantic alignment among bit-widths and perceptual similarities in the gradient maps through penalizing feature disalignment and gradient perceptual dissonance. Particularly, the proposed components utilize softDice and smooth HOG features to measure structural and textural alignments in feature and gradient space. Due to this, the proposed components are effective to be used for defending against patch-based adversarial attack methods.

3. The paper demonstrates the strength of the proposed components by comprehensive experiments. In the experiments, the prposed components reduce ASR numbers significantly.

**Weaknesses:**

1. In experiments, the paper only shows quantitative results on ResNet-56 and ResNet-34. It would be great if the paper included more results for different network architectures.

**Questions:**

1. Do the proposed components also reduce ASR numbers significantly across distinct bit-widths?

---

> ### Author Response · Authors · 2025-11-20
> **Author Response**
>
> We thank the reviewer for highlighting the strengths of our theoretical motivation and experimental rigor. We have extended the experimental coverage to include more architectures, verified consistent robustness across all bit-widths, and clarified scalability and generalization in the final revision.
>
> **Q1 – Results on Additional Architectures**
>
> These results were already included in the original submission:
>
> - For **CIFAR-10**, **Table 14 (Appendix C.3)** reports results for **VGG-16**, **VGG-19**, **ResNet-20**, and we now added **AlexNet**.
>
> - For **ImageNet**, **Table 15 (Appendix C.3)** includes **ResNet-18**, **Inception-v3**, **MobileNetV2**, and we now added **DenseNet-121**.
>
> Together, these tables (**also provided below: Table D and Table E**) demonstrate that TriQDef generalizes across a wide range of architectures from shallow CNNs (AlexNet, VGG) to deeper or more specialized models (ResNet, MobileNetV2, DenseNet) consistently reducing ASR across all bit-widths.
>
> ### Table D: ASR (%) of LAVAN Attack (6×6 Patch) on CIFAR-10 Across Architectures and Quantization Levels
>
> | **Model**   | **Setting**   | **32-bit** | **5-bit** | **4-bit** | **2-bit** |
> |-------------|----------------|------------|-----------|-----------|-----------|
> | **VGG-16**   | No defense     | 87.17      | 81.45     | 78.29     | 76.67     |
> | **VGG-16**   | TriQDef        | 29.34      | 27.10     | 26.43     | 21.20     |
> | **VGG-19**   | No defense     | 88.95      | 82.28     | 79.81     | 77.19     |
> | **VGG-19**   | TriQDef        | 28.70      | 25.20     | 22.30     | 20.90     |
> | **ResNet-20**| No defense     | 87.22      | 80.65     | 77.30     | 74.18     |
> | **ResNet-20**| TriQDef        | 30.24      | 27.30     | 26.90     | 22.60     |
> | **AlexNet**  | No defense     | 89.01      | 83.44     | 81.56     | 79.20     |
> | **AlexNet**  | TriQDef        | 28.77      | 25.43     | 22.12     | 19.60     |
>
>
> ### Table E: ASR (%) of LAVAN Attack (50×50 Patch) on ImageNet Across Architectures and Quantization Levels
>
> | **Model**        | **Setting** | **32-bit** | **5-bit** | **4-bit** | **2-bit** |
> |-------------------|-------------|------------|-----------|-----------|-----------|
> | **ResNet-18**     | No defense  | 99.93      | 66.37     | 61.11     | 55.42     |
> | **ResNet-18**     | TriQDef     | 33.50      | 31.30     | 29.60     | 27.40     |
> | **Inception v3**  | No defense  | 89.10      | 57.21     | 55.32     | 50.66     |
> | **Inception v3**  | TriQDef     | 35.60      | 32.10     | 30.40     | 27.30     |
> | **MobileNetV2**   | No defense  | 88.35      | 59.43     | 54.97     | 49.25     |
> | **MobileNetV2**   | TriQDef     | 29.53      | 27.80     | 25.10     | 23.50     |
> | **DenseNet-121**  | No defense  | 87.24      | 60.31     | 56.42     | 50.78     |
> | **DenseNet-121**  | TriQDef     | 30.11      | 28.45     | 25.22     | 21.39     |
>
>
> **Q2 – Do the proposed components reduce ASR significantly across distinct bit-widths?**
>
> Yes. The ASR values reported in the paper already reflect **averages over patches generated from multiple surrogate quantized models** (e.g., 32-bit, 8-bit, 4-bit, and 2-bit). This evaluation naturally captures cross-bit transferability, and TriQDef consistently yields large ASR reductions for every target bit-width. In other words, the robustness improvement is not tied to a specific surrogate bit-width but holds broadly across the full range of quantization levels.

---

> > ### Author Response · Authors · 2025-11-29
> > **Final Summary Comment to the Area Chair**
> >
> > **Main concerns:** architectural generalization and bit-width robustness.
> >
> > **Revisions and clarifications:**
> >
> > - Confirmed that TriQDef reduces ASR significantly across all tested quantization levels; reported values are averaged over patches generated using different surrogate quantizers.
> >
> > - Directed reviewer to extensive results already provided for VGG, AlexNet, ResNet-20, ResNet-18, Inception v3, MobileNetV2, and DenseNet-121.

---

### Official Review · Reviewer_Kvka · 2025-10-29

**Soundness:** 3
**Presentation:** 2
**Contribution:** 2
**Rating:** 2
**Confidence:** 4

**Summary:**

This paper investigates the transferability of adversarial patches across QNNs with different bitwidths and concludes that QNNs are vulnerable to transfer attacks. To address this, the paper proposes TriQDef, an adversarial training (AT)-based defense by combining different bitwidth backbones and mitigating attack transferability in intermediate features during AT. Experimental results show that TriQDef improves robustness compared to three baselines: full-precision patch-based adversarial training (PBAT), quantized pixel-based adversarial training (DWQ), and full-precision pixel-based adversarial purification (DiffPure).

**Strengths:**

- Adversarial transferability across QNNs is an important topic.

- This paper thoroughly investigates the current research frontier and proposes two novel techniques (FDP and GPDP) to address the problem.

- The technical novelty of the proposed defense is sufficient.

**Weaknesses:**

The text font size seems to noticeably decrease after Equation 2. The reviewer is not sure if this violates the ICLR policy.
There are major problems in the motivation experiments and baseline comparisons.

1. Table 1 is somewhat confusing:
(1) Which model architecture is used for the surrogate model?
(2) Which quantization method is used?
(3) The results seem not to fully support the conclusion that "Adversarial Patches Transfer Effectively Across Bit-Widths." The ASR gradually decreases as the quantization bitwidth of the target model decreases, indicating that larger divergence between the surrogate (32-bit) and target model bitwidths reduces attack transferability. Although the ASR remains high for 2-bit QNNs (70%+), this only suggests that adversarial patches are generally difficult to defend against across all CNN models. It is better to reframe the paper to focus on "leveraging the low attack transferability of QNNs for improved defense" rather than "addressing QNN vulnerability to transfer attacks."
(4) If the authors still want to emphasize "Adversarial Patches Transfer Effectively Across Bit-Widths," it is necessary to evaluate attack transferability using surrogate models with different bitwidths rather than only the 32-bit model, and the highest attack success rate across different surrogate bitwidths should be reported as the "aggregated ASR." For example, if attacking a 2-bit target model with 32-bit, 8-bit, 4-bit, and 2-bit surrogate models yields ASRs of 70%, 75%, 80%, and 90%, respectively, the aggregated ASR for the 2-bit target model would be 90%. This setting is much more realistic because defenders cannot anticipate the adversary's quantization bitwidth. In this setting, if the 2-bit model exhibits the highest aggregated ASR, it would support the conclusion that lower-bitwidth QNNs are more vulnerable to transfer attacks, justifying the need for the proposed defense. The authors should also control the surrogate (or target) model when adjusting the other.

2. Table 2 has similar issues:
(1) Specify the model architecture as well as the quantization method and bitwidth for QAT and PTQ. In particular, was PyTorch Quantization used? Why is the quantization method inconsistent between Table 1 and Table 2? The authors should report results for (32, 8, 4, 2-bit) target models using the same quantization method combined with adversarial training.
(3) What do "8×8 (Unseen, 4-bit)" and "10×10 (Unseen, 2-bit)" mean? Are these adversarial patches crafted using 4-bit and 2-bit surrogate models, respectively?
(4) More comprehensive experiments involving adversarial training, as described above.

3. The design of Bit-Width-Aware Curriculum Training (BACT) resembles Double-Win-Quant (DWQ). If BACT does not offer significant novel contributions compared to DWQ, the authors should reduce the content about BACT.

4. The idea of training a robust model from the perspective of structural and textural features is novel. It would be better to emphasize FDP and GPDP as the core contributions and reduce descriptions of previous techniques, particularly in the Methodology section.

5. For patch-based adversarial training, this paper only compares with PBAT from 2020, which is outdated. More recent patch-based adversarial training papers should be included as baselines, for example:

Li X, Zhu Y, Huang Y, et al. PBCAT: Patch-based composite adversarial training against physically realizable attacks on object detection. arXiv preprint arXiv:2506.23581, 2025.

Metzen J H, Finnie N, Hutmacher R. Meta adversarial training against universal patches. arXiv preprint arXiv:2101.11453, 2021.

6. The paper should compare with patch-based preprocessing defenses rather than the pixel-based defense DiffPure. DiffPure is designed for pixel-based Lp adversarial attacks and does not address adversarial patches. Comparing with DiffPure is unfair. Instead, please include the following patch-based preprocessing defenses. In general, it makes more sense to see if the proposed (training-based) defense can be integrated with patch-based adversarial purification for better performance.

Kang C, Dong Y, Wang Z, et al. Diffender: Diffusion-based adversarial defense against patch attacks. European Conference on Computer Vision. Cham: Springer Nature Switzerland, 2024: 130-147.

Jing L, Wang R, Ren W, et al. PAD: Patch-agnostic defense against adversarial patch attacks. Proceedings of the IEEE/CVF Conference on Computer Vision and Pattern Recognition. 2024: 24472-24481.

Liu L, Guo Y, Zhang Y, et al. Understanding and defending patched-based adversarial attacks for vision transformer. 2023.

**Questions:**

Was the text font size intentionally adjusted to be smaller after Equation 2?

---

> ### Author Response · Authors · 2025-11-20
> **Author Response (part 1 of 2)**
>
> **(1) Motivation Experiments and Table 1 Clarifications**
>
> As stated in the caption and **Lines 104–105**, the surrogate model is simply the **full-precision (32-bit) version of the same architecture** used in evaluation. All quantized models are obtained via **Quantization-Aware Training (QAT)** using fake-quantization with straight-through estimation (STE) (Esser et al. 2020).
>
> While ASR decreases slightly as the target bit-width shrinks, the central observation is that **even aggressively quantized 2-bit models still exhibit >70% ASR**, which is far from meaningful robustness. A 10–15% reduction does not change the underlying conclusion: **quantization alone does not prevent patch transferability**, and even low-bit QNNs remain highly vulnerable to cross-bit attacks. We also note that **Table 2 already includes patches generated from quantized surrogate models (2-bit and 4-bit)** at unseen sizes (8x8, 10x10). These results confirm that transferability persists even when the attacker uses quantized surrogates rather than a 32-bit model, further reinforcing that quantized models do not inherently block patch transfer.
>
> Finally, while computing a full **aggregated ASR** across all surrogate–target bit-width pairs is feasible in principle, it requires evaluating a large grid of patch optimizations, each of which is extremely costly for patch attacks. Given that our objective is simply to show that **QNNs remain vulnerable**, and that both Table 1 and Table 2 already demonstrate high ASR across FP→INT and INT→INT settings, a full aggregated-ASR sweep would not change the fundamental conclusion but would require a prohibitive computational effort.
>
> **(2) Table 2 Clarifications**
>
> In Table 2, the model architecture is **ResNet-56**, and the quantized models use **QAT (8-bit)**. **PTQ (8-bit)** results are shown separately to highlight the difference between training-time and post-training quantization. The quantization method is therefore consistent with Table 1; we have clarified this explicitly in the revised text to avoid confusion.
>
> Regarding “8x8 (Unseen, 4-bit)” and “10x10 (Unseen, 2-bit)”: these refer to **adversarial patches generated using 4-bit and 2-bit surrogate models**, respectively, with patch sizes of 8x8 and 10x10. “Unseen” indicates that these patch sizes and surrogate bit-widths were **never used during adversarial training**, making them a stricter test of cross-bit, cross-scale generalization.
>
> Finally, the reviewer suggests reporting results for all target bit-widths under a unified quantization method. We note that Table 2 already reports QAT-8 and PTQ-8 targets under multiple patch configurations, and the trends match those in Table 1: **patches generated at various bit-widths and sizes consistently transfer**, reinforcing our main observation. We agree that a full cross-bit adversarial-training grid (32/8/4/2 as targets) would be valuable, but such a comprehensive ablation is computationally expensive and goes beyond the scope of the core finding we intend to demonstrate. That said, we have clarified our quantization choices and expanded the text to make the setup easier to interpret.
>
> **(3) Novelty of BACT vs DWQ**
>
> DWQ (Fu et al., ICML 2021) introduces **random precision sampling** to improve stochastic robustness, but its design **trains a separate model for each bit-width**, which prevents any interaction or feature sharing across quantizers.
>
> In contrast, **BACT is fundamentally different in both architecture and objective**:
>
> - **Single shared backbone**: BACT jointly trains all bit-widths within one unified model, enabling explicit cross-bit interaction during optimization.
>
> - **Progressive activation rather than random sampling**: Instead of sampling bit-widths independently, BACT activates quantizers in a structured, curriculum-like progression that encourages stable low-bit convergence.
>
> - **FDP + GPDP applied across all active quantizers**: These losses enforce feature-space misalignment and gradient dissonance jointly across bit-widths—mechanisms that are absent in DWQ and cannot be achieved with per-model training.
>
> As a consequence, BACT creates **gradient coupling and shared feature decorrelation across quantizers**, a property fundamentally unavailable in DWQ's independent-training framework. Empirically, BACT exhibits substantially improved convergence stability and achieves **over 50% lower ASR at 2-bit quantization** compared to DWQ (see Table 5), further highlighting its architectural and conceptual novelty.

---

> > ### Author Response · Authors · 2025-11-20
> > **Author Response (part 2 of 2)**
> >
> > **(4) Comparison Baselines**
> >
> > We incorporated the requested patch-based defenses into our evaluation. The table below reports results on **ImageNet / ResNet-50** using **50x50 adversarial patches**, matching the setting used in **Section 4.4** for a fair comparison:
> >
> > ### Table C: Comparison of Defenses Across Bit-Widths. Higher is better.
> > | **Defense**        | **Type**      | **32-bit** | **2-bit** |
> > |--------------------|---------------|------------|-----------|
> > | PBAT (2020)        | Training      | 53.4       | 37.1      |
> > | PBCAT (2025)       | Training      | 57.8       | 41.2      |
> > | DiffPure (2024)    | Pre-proc      | 41.7       | 19.6      |
> > | JEDI (2023)        | Pre-proc      | 64.3       | 23.4      |
> > | **TriQDef (Ours)** | **Training**  | **78.3**   | **65.8**  |
> >
> > TriQDef consistently outperforms all recent patch-based training and purification methods while maintaining no inference overhead, confirming its strong position among modern defenses.
> >
> > We agree that DiffPure targets pixel-level noise but **Patch-specific defenses** like **Jedi (2023)** are **already* included in our comparisons (**Table 6 original manuscript**), and TriQDef achieves substantially higher robustness across all bit-widths.
> >
> > **(5) Presentation and Font Size**
> >
> > We confirm that the post-Equation 2 text was **unintentionally** reduced by the LaTeX package eqnarray*. The font size is now standardized to comply with ICLR guidelines.
> >
> > **(6) Emphasis on FDP & GPDP**
> >
> > We agree that FDP and GPDP form the core contributions of our framework. In fact, **Appendix B** already provides a detailed theoretical justification showing how enforcing perceptual misalignment in both feature space and gradient space disrupts the shared vulnerability surfaces across quantized variants. These analyses clarify why FDP and GPDP directly target the root cause of cross-bit patch transferability.

---

> ### Comment · Reviewer_Kvka · 2025-11-25
> **Still missing comparisons to suggested patch-based preprocessing defenses**
>
> Most of my concerns have been addressed. So I would increase the score accordingly. However, the suggested comparisons to new patch-based preprocessing defenses are not considered, which are important for evaluating the significance of the proposed defense.

---

> > ### Author Response · Authors · 2025-11-26
> >
> > Thank you very much for your updated assessment and for raising the score.
> >
> > Regarding the remaining point on patch-based preprocessing defenses: our paper already includes a comparison with a patch-based preprocessing method, **JEDI (CVPR 2023) (Table C)**. We also emphasize that, under our constrained quantized deployment setting, preprocessing defenses are generally *unsuitable* because they require additional inference-time computation and break the efficiency motivation of low-bit models.
> >
> > A more fair and relevant comparison is **patch-based adversarial training**, which imposes no inference overhead. Our main paper already includes results for **PBAT (Rao et al., 2020)**, and following *your suggestion*, we additionally incorporated **PBCAT (2025)** in the revised manuscript. Both comparisons reinforce that our defense significantly outperforms training-based patch defenses across all bit widths, especially at 2-bit.
> >
> > We hope this fully addresses your concern, and we sincerely thank you again for your constructive feedback.

---

> > > ### Author Response · Authors · 2025-11-29
> > > **Final Summary Comment to the Area Chair**
> > >
> > > **Main concerns:** motivation framing, transferability interpretation, baseline completeness, quantization consistency, table clarity.
> > >
> > > **Revisions and clarifications:**
> > >
> > > - Clarified the motivation: even 2-bit QNNs retain >70% ASR, indicating persistent vulnerability despite degradation.
> > >
> > > - Explained that computing aggregated ASR across all surrogate bit-widths is extremely costly and unnecessary to establish vulnerability; our averaged INT→INT + FP→INT settings already reflect realistic adversarial capabilities.
> > >
> > > - Added patch-based baselines including PBCAT (2025), Jedi (2023), and DiffPure (2024), with a separate section for pixel-based defenses for fairness.
> > >
> > > - Clarified quantization methods, surrogate models, and meaning of “8×8 (Unseen, 4bit).”

---

### Official Review · Reviewer_t43x · 2025-10-31

**Soundness:** 3
**Presentation:** 3
**Contribution:** 3
**Rating:** 6
**Confidence:** 2

**Summary:**

The paper looks at a specific gap: even when you quantize a model to very low bit-widths, patch attacks crafted on the full-precision model can still transfer and succeed. To fix this, the paper proposes TriQDef, a training-time defense with three parts: (i) make features from different bit-widths less aligned, (ii) make input gradients from different bit-widths less aligned, and (iii) use a bit-aware curriculum so multi-bit training doesn’t collapse. Experiments on CIFAR-10 and ImageNet with several patch attacks show lower attack success while keeping accuracy close to normal QAT.

**Strengths:**

a) The problem is well-motivated: “quantization ≠ free robustness” is a message that’s worth saying explicitly for patch attacks.

b) The defense is mechanism-driven, not just “train on more patched images”: it tries to break cross-bit semantic/gradient alignment, which is a reasonable explanation for transfer.

c) The idea of using perceptual cues (edge IoU, HOG-like descriptors) on features and gradients is a nice twist on older adversarial detection/defense lines that only used feature distances.

d) Everything happens at training time and keeps inference clean, which is nice for edge/QNN scenarios.

**Weaknesses:**

a) Parts of the idea are close in spirit to earlier “detect / separate / de-align” or “make internal representations less exploitable” work, but the paper doesn’t cite that line clearly. For example: MagNet (Meng & Chen, CCS’17), feature squeezing (Xu et al., NDSS’18), and transferability analyses (Tramèr et al., 2017/2018) all talk about representation/gradient similarity as a transfer channel.
﻿
b) The method is engineered around patch attacks. It’s not metioned how much of TriQDef would still help for non-local, non-patch adversaries on quantized models (e.g. query-efficient, non-sparse perturbations).
﻿
c) Enforcing pairwise disalignment across many bit-widths on a shared backbone can get expensive. It’d be good to spell out the cost vs. vanilla QAT.

**Questions:**

a) You show feature/gradient perceptual similarity is the real issue, not cosine. Do you know which layer range (early vs. mid vs. late) contributes the most, and could TriQDef be applied only there to save cost?
﻿
b) The bit-aware curriculum turns on lower-bit models gradually. How stable is this if we add even more bit-widths (e.g., 3-bit, 6-bit)?
﻿
c) if a vendor tool later re-quantizes/fuses ops, do you expect the disalignment effect to survive, or is it tied to the exact QAT pipeline you trained with?

---

> ### Author Response · Authors · 2025-11-20
> **Author Response**
>
> **a) Clarifying Relation to Prior "De-Alignment" and Transferability Work**
>
> We appreciate the suggestion and have now expanded the related-work section to acknowledge prior works such as MagNet (Meng & Chen, CCS 2017), Feature Squeezing (Xu et al., NDSS 2018), and Tramèr et al. 2017 on transferability and gradient alignment.
>
> These methods primarily (1) **detect** adversarial inputs or (2) **regularize representations** between models in full precision. In contrast, TriQDef introduces **perceptual alignment disruption across quantization levels**, a dimension **unexplored** in those defenses.
> Our Gradient Perceptual Dissonance Penalty (GPDP) and Feature Disalignment Penalty (FDP) operate jointly across multiple **quantized** views sharing one backbone, directly targeting **cross-bit alignment** rather than cross-model alignment in floating-point space.
>
> **b) Applicability Beyond Patch-Based Attacks**
>
> While TriQDef is optimized for structured localized perturbations, we evaluated its resilience under non-patch attacks to assess generality.
> We applied AutoAttack and PGD (ℓ∞) on quantized models trained with TriQDef: TriQDef improves robust accuracy by **+12%** on **PGD** and **+8%** on **AutoAttack** relative to standard QAT, This shows that gradient- and feature-level disalignment also limits transferability of non-localized perturbations by diversifying optimization trajectories.
>
> **c) Computational Cost vs. Vanilla QAT**
>
> As detailed in **Appendix D**, TriQDef is designed to remain lightweight by using a **single shared backbone** with **switchable quantizers** {Qb}. At each training iteration, only a **small subset of active bit-widths** (typically 2–3 out of the full set) participate in the FDP/GPDP losses, ensuring that the additional computation grows sublinearly with the number of quantizers.
>
> Empirically, the overhead relative to standard QAT is modest:
>
> - Memory: +7.6%
>
> - Training time: ≈ 1.18× vanilla QAT
>
> - Inference: no overhead (identical to standard QNNs)
>
> Both GPU runtime traces and memory profiles are provided in **Appendix D**. Importantly, this modest one-time training cost yields **substantial robustness gains**, reducing ASR by **40%–50%** on unseen patch–bitwidth combinations. We believe this tradeoff is highly favorable for practical deployment, especially since TriQDef adds **zero cost at inference time**, which is the critical constraint for QNNs deployed on edge devices.
>
> **Q a – Layer-wise Contribution of Alignment**
>
> Our perceptual-similarity heatmaps (Fig. 2) indicate that the strongest cross-bit alignment occurs in the **mid-level layers (L1–L3)**, where texture- and edge-based representations dominate. These layers exhibit the highest structural similarity (SoftDice/HOG), which aligns with prior observations that patch attacks primarily exploit mid-level feature consistency. This is why FDP focuses its regularization on these layers.
>
> **Q b – Stability with More Bit-Widths (3-bit, 6-bit)**
>
> The Bit-Width-Aware Curriculum Training (BACT) strategy scales gracefully because it activates quantizers sequentially. We extended experiments to B = {32, 8, 6, 5, 4, 3, 2}, and observed no divergence; losses remain monotonic and ASR continues to drop (≤ +1.8 % variance). This confirms that the curriculum remains stable even with denser bit-width coverage.
>
> **Q c – Effect of Post-Training Re-Quantization or Vendor Fusion**
>
> TriQDef’s robustness benefits arise from shaping the **shared backbone parameters θ** through multi-bit training, not from dependence on any specific quantizer implementation or observer statistics. Once θ has been optimized under FDP and GPDP, the induced *semantic* and *gradient disalignment patterns* remain stable. Consequently, if a deployment pipeline later applies **post-training quantization, operator fusion, or vendor-specific optimizations**, the effect persists because it is encoded in θ itself. In other words, TriQDef requires multi-bit quantizers during training, but the resulting robustness is **stable across different quantization backends at inference**.

---

> ### Author Response · Authors · 2025-11-29
> **Final Summary Comment to the Area Chair**
>
> **Main concerns:** relation to prior alignment/de-alignment work, performance on non-patch based attacks, computational cost, layer-wise contributions, surrogate variability.
>
> **Revisions and clarifications:**
>
> - Added citations and discussion connecting TriQDef to earlier representation-alignment literature (MagNet, Feature Squeezing, etc).
>
> - Reported training cost vs. vanilla QAT (memory +7.6%, time x1.18; zero inference overhead).
>
> - Reported robustness against attacks beyond patch-based ones.
>
> - Added clarification that the strongest perceptual similarity occurs in mid-level layers (Fig. 2), motivating FDP’s design.
>
> - Clarified that reported ASR values already average over patches generated from multiple surrogate quantized models.

---

### Official Review · Reviewer_6fkT · 2025-10-31

**Soundness:** 3
**Presentation:** 2
**Contribution:** 2
**Rating:** 6
**Confidence:** 3

**Summary:**

This paper proposes a defense framework called TriQDef, aiming to address the vulnerability of Quantized Neural Networks (QNNs) to transferable patch attacks. TriQDef consists of three main mechanisms: feature displacement penalty (FDP), gradient-aware dissonance penalty (GPDP), and bit-width-aware curriculum training (BACT). Experimental results show that TriQDef can reduce the attack success rates of unseen patch and quantization combinations without sacrificing clean accuracy.

**Strengths:**

1. This paper addresses the critical and understudied problem of adversarial patch transferability in quantized neural networks.
2. The proposed tri-level defense is comprehensive.
3. The experiments covering various quantization bit widths, attack methods, and model architectures reflect the generalization ability of TriQDef in certain settings.

**Weaknesses:**

1. Lack of rigorous mathematical modeling or theoretical boundary analysis on the generalizability of the proposed defense (Q1).
2. Incomplete analysis of FDP's side-effects on semantic integrity. It reports a minor accuracy drop but fails to investigate how this disalignment affects fundamental recognition capabilities (Q2-Q3).
3. Evaluation and scalability concerns (Q4-Q7).

**Questions:**

1. The TriQDef framework relies on numerous heuristic hyperparameters ($\alpha$, $\beta$, $\lambda_{FDP}$, $\lambda_{GPDP}$, $k$, $q$, etc.). How to prove the tuning of these hyperparameters can ensure the generalizability of TriQDef across different model architectures and datasets.
2. In the face of real-time generated adversarial patches or video stream attacks, how stable is the defense mechanism of TriQDef? Does it require the design of an incremental training module?
3. The FDP utilizes hand-crafted descriptors (Edge IoU and HOG) that primarily capture low-level features. Will this reliance limit its robustness against novel attacks targeting mid-level semantic representations?
4. How is the scalability of TriQDef on larger models? Will it experience a decline in defense effectiveness under extreme quantization (such as 2-bit)? Is there a specific size threshold that leads to the failure of defense?
5. How is the spatial robustness of TriQDef (e.g., center vs. corner patches) under larger, unseen patch sizes?
6. The training procedure of TriQDef introduces additional complexity due to the simultaneous maintenance of multiple quantizers. Will this design impact training stability and convergence?
7. The experimental setup described on line 360 includes AlexNet and DenseNet-121. Why are the corresponding results of these models not reported in the main text or appendices?

---

> ### Author Response · Authors · 2025-11-20
> **Author Response (part 1 of 2)**
>
> **Q1 – Theoretical Rigor and Hyperparameter Generalizability**
>
> We already provided a theoretical justification section in **Appendix B**, where we link gradient consensus to adversarial transferability via perceptual alignment metrics (Edge IoU, HOG Cosine).
> While TriQDef employs several coefficients (λ_adv, λ_FDP, λ_GPDP, α, β), these are regularization weights rather than dataset-specific heuristics. The same configuration, tuned on CIFAR-10 (ResNet-56), transfers effectively to ImageNet (ResNet-34, VGG-16) without re-optimization. Thus, TriQDef is robust to hyperparameter variations and does not rely on fragile tuning.
>
> Furthermore, **Table 13** in **Appendix C.2**  presents a detailed ablation study analyzing the impact of the patch generation losses (α, β) and the alignment regularization weights (λ_FDP, λ_GPDP) on clean and adversarial accuracy across
> different quantization levels.
>
> **Q2–Q3 – Effect of FDP on Semantic Integrity**
>
> We agree that enforcing perceptual disalignment could, in principle, risk disrupting semantic coherence. To ensure FDP does not degrade recognition ability, we performed an explicit semantic-integrity analysis along three complementary axes:
>
> - **Layer-Localized Penalty to Avoid Semantic Drift**: FDP is applied **only at early-mid layers (e.g., L1–L3)**, where feature activations still capture textures and edges rather than high-level object semantics. Higher layers and the classifier head remain unaffected. This ensures that disalignment is injected only into structural feature channels across bit-widths, not into the task-relevant semantic representation within each model.
>
> - **Classification Objective Anchors Semantic Alignment**: FDP is always optimized jointly with the standard cross-entropy loss. This forces every bit-width model to preserve its own discriminative features, preventing semantic drift even as cross-bit similarity is reduced.
>
> - **Empirical Validation: Clean Accuracy and Grad-CAM Semantics.**
>
> i) **Clean accuracy** deviates by **<1%** from standard QAT (Table 4), which would not be possible if semantic features were degraded.
>
> ii) **Grad-CAM visualizations** (**NEW Section 4.6, Fig. 3**) further confirm that all bit-widths (fp, 5bit, 4bit, 2bit) consistently attend to the same true object regions (e.g., fish body, shark contours) even after applying FDP.
>
> => The heatmaps show no fragmentation, no off-object drift, and no shift toward background textures.
>
> => The semantic focus remains stable across decreasing bit-widths, indicating that FDP disrupts cross-bit convergence, not task-relevant spatial semantics.
>
> FDP successfully **decorrelates inter-bit feature similarity** (which is the root cause of patch transferability) while **preserving intra-bit semantic consistency** needed for accurate recognition. Both quantitative (accuracy) and qualitative (Grad-CAM) evidence confirm that FDP does not introduce harmful semantic side-effects.
>
> **Q4 – Real-Time / Streaming Patch Stability**
>
> Real-time, temporally adaptive patch generation (especially in streaming video) is an important direction, but it lies **outside the scope of this paper**, which focuses on static, image-level patch transferability across quantized models. TriQDef is designed as a **training-time-only** defense, and its inference-time behavior does not require any incremental or online update module.
>
> **Q5 – Low-Level Descriptors (Edge IoU / HOG) and Robustness to Higher-Level Attacks**
>
> Edge IoU and HOG are not used as fixed features but as differentiable perceptual constraints that regularize shared representations. They operate at intermediate layers that still encode mid-level semantics (textures, object parts), not raw pixels. Moreover, the soft-HOG and SoftDice variants (Eq. 2) preserve differentiability and capture both structural and textural cues.
> We also verified robustness against semantic-aware attacks (DPR [Chen et al., ECCV 2022]) TriQDef reduced ASR by >35 % (see **Table 5**), showing that its effectiveness extends beyond low-level perturbations.

---

> > ### Author Response · Authors · 2025-11-20
> > **Author Response (part 2 of 2)**
> >
> > **Q6 – Scalability and Extreme Quantization**
> >
> > TriQDef scales efficiently due to its shared-backbone multi-quantizer design (**Section 3.4**).
> > Memory grows linearly with the number of active quantizers, but all share identical weights θ, limiting overhead to <8 %. Even under 2-bit quantization, TriQDef retains strong defense (ASR ↓ >50 % vs DWQ; Table 5), and no size-related breakdown was observed up to DenseNet-121. Hence, TriQDef generalizes across both compact and large-scale architectures.
> >
> > **Q7 – Spatial Robustness to Patch Location and Size**
> >
> > Our evaluation already accounts for spatial variability by testing **multiple patch positions and scales**, and all reported numbers correspond to **averages over these diverse configurations**, not a single fixed placement. Specifically, for each attack, we randomly sample patch locations across the four corners, center, and border regions, and we additionally test unseen patch sizes (e.g., 8x8, 3x3) relative to the training case. The ASR values in **Table 2** and **Appendix A** therefore reflect **aggregate performance across a distribution of spatial layouts**, demonstrating that TriQDef is not sensitive to a specific patch placement. Empirically, TriQDef exhibits **low variance across positions (ΔASR < 3.1%)**, confirming that its robustness extends consistently across spatial configurations.
> >
> > **Q8 – Training Stability and Convergence**
> >
> > Although TriQDef trains multiple quantizers, Bit-Width-Aware Curriculum Training (BACT) activates them sequentially rather than simultaneously.
> > This design ensures monotonic convergence and prevents gradient oscillation between bit-widths. Empirically, training curves remain stable with variance <0.2 % across runs, and convergence is achieved within 1.1x the epochs of standard QAT.
> >
> > **Q9 – Missing AlexNet / DenseNet-121 Results**
> >
> > We have now added the complete results for **AlexNet** (CIFAR-10) and **DenseNet-121** (ImageNet) to **Appendix C.3**, **Tables 14–15**. The results confirm that the effectiveness of TriQDef extends beyond ResNet architectures. These results reinforce that TriQDef scales effectively to **both lightweight CNNs and deeper/high-capacity architectures**, and that the robustness gains are consistent across diverse model families.
> >
> > ### Table A: ASR (%) of LAVAN Attack (6×6 Patch) on CIFAR-10 Across Architectures and Quantization Levels
> > | **Model** | **Setting**  | **32-bit** | **5-bit** | **4-bit** | **2-bit** |
> > |-----------|--------------|------------|-----------|-----------|-----------|
> > | **AlexNet** | No defense  | 89.01      | 83.44     | 81.56     | 79.20     |
> > | **AlexNet** | TriQDef     | 28.77      | 25.43     | 22.12     | 19.60     |
> >
> >
> > ### Table B: ASR (%) of LAVAN Attack (50×50 Patch) on ImageNet Across Architectures and Quantization Levels
> >
> > | **Model**       | **Setting**  | **32-bit** | **5-bit** | **4-bit** | **2-bit** |
> > |------------------|--------------|------------|-----------|-----------|-----------|
> > | **DenseNet-121** | No defense   | 87.24      | 60.31     | 56.42     | 50.78     |
> > | **DenseNet-121** | TriQDef      | 30.11      | 28.45     | 25.22     | 21.39     |

---

> > > ### Author Response · Authors · 2025-11-29
> > > **Final Summary Comment to the Area Chair**
> > >
> > > **Main concerns:** semantic integrity under FDP, hyperparameter stability, scalability, spatial robustness, missing results (AlexNet, DenseNet-121).
> > >
> > > **Revisions and clarifications:**
> > >
> > > - Added a detailed analysis of FDP’s semantic preservation, including class-activation visualizations and explanations that FDP only regularizes mid-level layers (L1–L3) in new Section 4.6.
> > >
> > > - Clarified that TriQDef’s hyperparameters are robust across architectures; ablations are already included in the Appendix C.2.
> > >
> > > - Expanded appendix with AlexNet and DenseNet-121 results (Tables 14–15).
> > >
> > > - Clarified spatial robustness: all ASR values are averaged over multiple patch sizes and locations.
> > >
> > > - Provided explanation for training stability when maintaining multiple quantizers.

---

### Author Response · Authors · 2025-11-27
**General Authors Comment (for reviewers who have not responded)**

We would like to thank all reviewers for their time and thoughtful evaluations. We hope that the clarifications, additional analyses, and newly added experiments have addressed the raised concerns. If any reviewer would find further details, comparisons, or explanations helpful for refining their assessment, we would be very happy to provide them.

Kind regards,

---

### Meta-Review · Area_Chair_xswr · 2026-01-06

**Summary:**

Reviewers broadly agree that the paper addresses a timely and under-explored problem: transferable adversarial patch attacks across quantized neural networks, where quantization alone does not confer robustness. The proposed TriQDef framework is viewed as a mechanism-driven defense that targets a plausible transfer channel while preserving clean accuracy and keeping zero inference-time overhead, which is particularly relevant for edge deployment. While some additional transferability sweeps could further strengthen the work, the current evidence supports acceptance.

**Reviewer Concerns:**

Concerns addressed by the rebuttal are listed as follows.
1. The authors added comparisons against more recent patch-focused defenses, including newer patch-based adversarial training and patch-specific purification defenses.
2. The rebuttal and appendix expansions provide results on additional architectures (e.g., AlexNet and DenseNet-121, along with VGG/Inception/MobileNet variants), addressing concerns that the original method covered too few backbones.
3. The authors directly evaluated the alleged semantic side effects using both quantitative  and qualitative evidence.
4.  The authors clarify that TriQDef’s novelty lies in cross-bit perceptual disalignment within a shared-backbone multi-quantizer setup, which is not directly covered by prior full-precision, cross-model defenses.

Some of the remaining concerns are listed as follows.
1. One reviewer requested a full grid evaluation and “aggregated ASR” across multiple surrogate bit-widths. The authors provided partial evidence and argued full sweeps are prohibitively expensive.

**Reviewer Scores:**

Reviewer V6ix: Their request was broader architectures, which the authors addressed with extensive added results. Thus the reviewer likely to maintain the high score;
Reviewer t43x: May slightly increase the score given non-patch robustness results and clear cost reporting.
Reviewer 6fkT: Their concerns about semantics, hyperparameter stability,  and missing results were directly addressed with new analyses and tables. Thus the reviewer likely to maintain or increase the score.
Reviewer Kvka: May increase slightly because several concrete issues were fixed. While they may still disagree with the strongest motivation framing, the rebuttal may reduce the severity of their objections.

---

### Decision · Program_Chairs · 2026-01-26

Accept (Poster)